# Folic acid supplementation during preconception period in sub-Saharan African countries: A systematic review and meta-analysis

**Mekuriaw Nibret Aweke**[1]*, **Elsa Awoke Fentie**[2], **Muluken Chanie Agimas**[3], **Lemlem Daniel Baffa**[1], **Ever Siyoum Shewarega**[4], **Aysheshim Kassahun Belew**[1], **Esmael Ali Muhammad**[1], **Berhanu Mengistu**[1]

1 Department of Human Nutrition, Institute of Public Health, College of Medicine and Health Sciences, University of Gondar, Gondar, Ethiopia, 2 Department of Reproductive Health, Institute of Public Health, College of Medicine and Health Science, University of Gondar, Gondar, Ethiopia, 3 Department of Epidemiology and Biostatistics, Institute of Public Health, College of Medicine and Health Science, University of Gondar, Gondar, Ethiopia, 4 Department of Reproductive Health, School of Public Health, College of Medicine and Health Science, Dilla University, Dilla, Ethiopia

* mekunib@gmail.com

**Data Availability Statement:** The data used in this study are from the Ethiopian Mini Demographic and Health Survey (miniDHS) 2019 and can be

## Abstract

### Introduction

Neural tube defects (NTDs) are serious congenital anomalies of the central nervous system caused by disruptions in early embryonic development. The prevalence is about twice as common in low- and middle-income countries and more prevalent in sub-Saharan Africa. Folic acid deficiency is a major risk factor and common during pregnancy. However, limited research on preconception folic acid supplementation in this region highlights the need for systematic reviews and targeted interventions to improve maternal and fetal health.

### Methods

We conducted a systematic literature search in EMBASE, MEDLINE, Scopus, CINAHL, Google Scholar, and Google for studies on the proportion of folic acid supplementation during the preconception period, covering publications up to January 2024. Study quality was assessed using the Newcastle-Ottawa Scale. Heterogeneity was evaluated with Cochrane Q and $I^2$ statistics, and small study effects were tested with Egger's test at a 5% significance level. The certainty of evidence was assessed using the GRADE approach. A random-effects model was used to estimate the pooled proportion of FA supplementation during the preconception period in sub-Saharan African countries.

### Result

This systematic review included 28 studies with a total of 12,562 participants. The highest (45.2%) and lowest (1.9%) proportion of folic acid supplementation during preconception period were recorded in the southern and Amhara regions of Ethiopia, respectively. The

requested from the DHS Program at https://dhsprogram.com/Data/ following the procedures outlined in the Materials and Methods section of the paper. Access to these data is granted upon registration and approval by the DHS Program.

**Funding:** The author(s) received no specific funding for this work.

**Competing interests:** The authors have declared that no competing interests exist.

**Abbreviations:** CI, Confidence Interval; FA, Folic Acid; PRISMA, Preferred Reporting System for Meta-Analysis and Systematic Review; NTDs, Neural Tube Defects; SSA, Sub-Saharan Africa; WHO, World Health Organization.

estimated pooled proportion of folic acid supplementation among women in Sub-Saharan Africa (SSA) during preconception period was (14.10%; 95% CI: 11.22–16.98) with significant heterogeneity between studies ($I^2$ = 97.71%, $p$ = 0.001). In sub-group analysis based on corresponding countries the highest estimated folic acid supplementation proportion during preconception period was found in studies conducted in Kenya ((22%; 95% CI: 19%-25%), $I^2$ = 97.7%), followed by studies conducted in Ghana (20%; 95% CI: 7%-33%), $I^2$ = 96.9%). The majority of the studies included in the analysis are of high quality, with quality scores ranging from 7 to 8. The certainty of evidence was assessed using the GRADE approach, resulting in a low overall rating.

## Conclusion

The results of this systematic review and meta-analysis indicated that folic acid supplementation during preconception period is significantly low among mothers in sub-Saharan African countries, despite being one of the best approaches to improve birth outcomes. Therefore, healthcare organizations, governments, policymakers, and other stakeholders involved in folic acid supplementation must collaborate on developing strategies to improve its uptake during the preconception period.

## 1. Introduction

Neural tube defects (NTDs) are serious congenital anomalies that affect the central nervous system, arising from disruptions in the early embryonic development process responsible for closing the neural tube [1]. Various types of NTDs, including anencephaly, spina bifida, encephalocele, craniorachischisis, and iniencephaly, cause significant mortality, morbidity, long-term disability, and substantial psychological and economic impacts [2,3].

Globally, neural tube defects (NTDs) occur in about 1 in 1,000 pregnancies, making them the second most common congenital anomaly. Annually, they result in 300,000 affected births, 88,000 deaths, and 8.6 million disability-adjusted life years (DALYs) [4–6]. The burden of NTDs has been documented to be approximately twice as high in low- and middle-income countries (LMICs) compared to high-income countries [7].

Various studies conducted in LMICs have found that the prevalence of NTDs ranges from 11.7 to 50.74 per 10,000 births, highlighting a significant burden on different segments of the population [8,9]. In sub-Saharan Africa (SSA), NTDs are prevalent congenital anomalies, impacting around 1 to 3 births per 1000 annually, with an overall prevalence of about 21.4% [10]. A systematic review of 58 studies across 16 African countries found the NTD burden in Africa was 32.95 per 10,000 births, with Eastern Africa at 111.13 and Southern Africa at 11.43 per 10,000 births [11]. Another meta-analysis in Africa reported a pooled prevalence of 50.71 per 10,000 births, highlighting the impact of folic acid (FA) supplementation and maternal exposures on NTDs in the region [9]. Fetuses with anencephaly usually die either before birth or shortly thereafter [12]. Neural tube defects (NTDs) impose a significant economic burden. For instance, the CDC estimated the lifetime cost of spina bifida care at $791,900 in 2014 [13].

The etiology of NTDs is not fully understood, but factors such as genetic predisposition, maternal nutrition, environmental influences, and insufficient folic acid intake during early pregnancy are implicated [14]. Folic acid insufficiency is an important risk factor for NTDs and is common during pregnancy because dietary intake is insufficient to cover daily needs,

even with a balanced diet [15]. Folate insufficiency exceeds 40% in many countries with available data, particularly affecting several African nations [16].

Multiple studies have highlighted that most NTDs can be prevented through effective FA supplementation during preconception [17]. Epidemiological evidence indicates that FA supplementation during preconception period reduces the incidence of NTDs when continued for a minimum of three months during pregnancy [18].

Folate is a water-soluble vitamin that is important in DNA synthesis, cell replication, and growth [19]. There are two strategies for improving FA intake prior to pregnancy: one involves fortifying the general food supply with folate, while the other entails targeted folate supplementation for individuals [20]. The WHO recommends women of reproductive age take 0.4 mg of folic acid daily, starting 2–3 months before conception and continuing through the 12th week of pregnancy [21,22]. High doses (4–5 mg per day of FA) are commonly recommended for those with certain underlying risk factors for NTDs and can reduce the risk of recurrent NTDs by more than 70 percent [21]. An alternative recommendation suggests weekly iron and folic acid supplements for women of reproductive age, especially in populations with anemia rates over 20% [23].

Folic acid supplementation is recommended before conception because NTDs occur very early in pregnancy, often before a woman knows she is pregnant [24]. Therefore, it is crucial to have adequate FA levels in the system before becoming pregnant. Since most pregnancies are unplanned, many countries have implemented FA fortification of staple foods for the prevention of NTDs [25].

Folic acid supplementation not only prevents NTDs but also reduces the occurrence of congenital cardiac and urologic anomalies [26]. Adequate folic acid intake before conception and through the 12th week of pregnancy is linked to reduced spontaneous abortions, preterm births, small-for-gestational-age babies, and improved pregnancy outcomes [27]. Despite established recommendations and benefits, folic acid supplementation before conception remains suboptimal, particularly in resource-limited settings [28]. The use of guidelines and recommendations for preconceptional FA supplementation varies globally, leading to differences in supplementation practices [29]. Several studies have indicated that taking FA before pregnancy is crucial for the health of both the mother and the fetus, particularly SSA countries [30]. There is a significant gap in the literature regarding folic acid supplementation during the preconception period in Sub-Saharan Africa. This systematic review and meta-analysis aims to estimate its proportion, serving as a crucial indicator for governments, health professionals, and policymakers to monitor progress, implement targeted interventions, and reduce the burden of NTDs at both individual and national levels.

## 2. Methods

### 2.1 Study protocol and registration

This systematic review and meta-analysis was reported according to the Preferred Reporting Items for Systematic Reviews and Meta-Analyses statement guideline [31] and the study protocol was registered in the PROSPERO International Prospective Register of Systematic Reviews under the registration number CRD42024503924.

### 2.2 Inclusion and exclusion criteria

In this systematic review and meta-analysis, we included all studies conducted on the proportion of FA supplementations during the preconception period in SSA countries that were written in English. Quantitative studies, including cross-sectional, cohort, and case-control designs, were included based on the Population, Exposure, Comparator, Outcome (PECO)

**Table 1. Criteria for inclusion of quantitative studies based on the Population, Exposure, Comparator, Outcome (PECO) framework.**

| Criteria | Details |
|---|---|
| Population (P) | Women of all age group in Sub-Saharan African (SSA) countries<br>Excludes women with diabetes, epilepsy, or prior neural tube defect-affected births |
| Exposure (E) | folic acid (FA) supplementation during the preconception period |
| Comparator (C) | Not applicable |
| Outcome (O) | Proportion of folic acid supplementation during the preconception period |

framework (Table 1). We included journal articles, theses, and dissertations without imposing restrictions based on publication date or age criteria. In contrast, studies utilizing qualitative methodologies were excluded due to the specific nature of the review and the chosen analytical approach. Studies done on pregnant women were only included if they had taken folate during preconception.

Exclusion criteria encompassed studies lacking data on the percentage of women engaging in any FA supplementation specifically during the preconception period as well as studies reporting zero prevalence. Additionally, experimental, case report or case series studies were excluded, along with studies where reported data were inadequate for estimating proportion (S1 Appendix). Furthermore, studies focusing on women with diabetes, epilepsy, or prior neural tube defect-affected births were excluded due to differing FA recommendations and potentially closer monitoring by healthcare providers during pregnancy.

## 2.3. Search strategy

The search strategy for this review involved using both MeSH terms and key terms related to preconception care and folic acid such as preconception care, preconception, pre-conception, periconception, peri-conception, before conception, folic acid, folate, vitamin, pteroylglutamic acid, folic acid supplementation, folic acid administration, folic acid adherence, folinic acid and vitamin B9. Terms were combined by using the Boolean operator OR and themes were combined using AND as explained in (S1 Table). Studies published in the English language up to January 2024 were retrieved from EMBASE, MEDLINE, Scopus, CINAHL, and manually on Google and Google Scholar. The search for unpublished studies included Google and institutional repositories(namely Addis Ababa University, Jimma University, Hawassa University, Haramaya University and Bahir Dar University). The literature search was last conducted on January 29, 2024 ensuring that the most recent and relevant studies were considered.

## 2.4. Study selection

We reviewed and evaluated studies from various sources to ensure a comprehensive analysis. All duplicate studies were removed, and full-text studies were downloaded. All the studies reviewed through different electronic databases were combined, exported, and managed using RAYYAN web-based software [32]. Titles and abstracts were first screened based on specific criteria to exclude irrelevant studies. Those that passed this stage underwent full-text screening, where their design, methodology, sample size, and key findings were assessed for relevance and suitability. The eligibility of each study was independently evaluated by two reviewers (BM & EA). Any significant differences in their assessments were resolved through discussion and consultation with other reviewers (MN & AK). Subsequently, the data were exported to the EndNote reference manager to facilitate the management and organization of citations.

## 2.5 Data extraction

The selected studies were carefully reviewed, and data were extracted using a pre-piloted data extraction form developed by the authors. Subsequently, data extraction was carried out using Microsoft Office Excel. All-important data extracted from each study included name of author, year of publication, study design, sub-Africa region, study period, country, study setting, publication type, sampling method(random/non-random), population characteristics, response rate, sample size (N) and overall proportion (S2 Appendix). We contacted one study author to obtain missing data to ensure comprehensive analysis. Data extraction was performed independently by three authors (E.S., A.K., and L.D.). Any discrepancies or inconsistencies were resolved through discussion and consensus.

## 2.6 Assessment of the quality of the included studies

Three reviewers (M.N., B.M and M.C.) assessed the methodological quality and eligibility of the identified articles using the Newcastle-Ottawa Quality Assessment Scale adapted for cross-sectional studies [33]. Disagreements among reviewers were resolved through discussion. This assessment scale evaluates several factors, including the representativeness of the sample, sample size adequacy, non-response rate, validity of measurement tools, comparability of subjects in different outcome groups, outcome assessment, and statistical testing. The quality assessment score for each study ranges from 0 to 10, with classifications including "Very Good" (9–10 points), "Good" (7–8 points), "Satisfactory" (5–6 points) and "Unsatisfactory" (0 to 4 points).

## 2.7 Evidence certainty assessment

In accordance with Cochrane's guidelines, we evaluated the certainty of the evidence related to preconception FA supplementation, synthesizing findings from the included studies. This evaluation utilized the GRADE (Grading of Recommendations, Assessment, Development, and Evaluation) framework, considering factors such as study design, inconsistency, indirectness, imprecision, and potential publication bias, as outlined in the GRADE handbook [34]. The assessment was adapted to proportion estimates, with the certainty of evidence categorized as high, moderate, low, or very low. The results were presented in a Summary of Findings (SoF) table, generated using the GRADE approach [35]. The certainty of evidence was assessed independently by E.A. and M.C., with any disagreements resolved through consensus.

## 2.8 Statistical analysis

The extracted data were transferred to STATA/MP version 17.0 software (Stata Corp LLC, College Station, TX, USA) for analysis. The pooled proportion of FA during preconception period was analyzed using the random effects model with DerSimonian-Laird model weight. Heterogeneity in meta-analysis is commonly inevitable due to variations in study quality, sample size, methodology, and outcome measurements across studies. Statistical assessment of significant heterogeneity was conducted using the Cochrane Q-test and $I^2$ statistics. The interpretation of the $I^2$ test statistics $< 50$, 50–75%, and $> 75\%$ was declared as low, moderate, and high heterogeneity, respectively [36]. To reduce variance in estimated points between primary studies, subgroup analyses were performed based on sub-Africa regions, countries, year of publication, study setting, sampling methods, and study population (various articles include pregnant individuals, others encompass mothers who have delivered, while some incorporate women in their childbearing years). Also, sensitivity analyses were done to assess the influence of included studies on pooled estimates. Publication bias was evaluated both graphically and using Egger's statistical test. A statistically significant result from Egger's test (P-value $< 0.05$)

would indicate the presence of a small study effect, which would then be addressed using non-parametric trim and fill analysis with the random effects model. We then proceed to explore the origins of heterogeneity using a meta-regression model, incorporating publication year and sample size as covariates.

## 3. Results

### 3.1 Selection and Identification of studies

Through a comprehensive and systematic exploration of literature, this systematic literature search identified 1,259 potential studies through electronic search and other sources. After eliminating the duplicates (n = 327), we reviewed 929 studies by title and abstract and eliminated 734 irrelevant studies. We reviewed the full texts of the remaining 195 articles for eligibility. Of these, 166 studies were not eligible, and 29 studies were eligible for inclusion in our review. After assessing the full texts of the remaining articles, an additional 1 article was excluded due to poor quality.

Finally, 28 studies that met the eligibility criteria were included in the final analysis (Fig 1).

### 3.2 Characteristics of included studies

This systematic review included 28 studies with a total of 12,562 participants [7,37–64]. All the studies utilized in this meta-analysis were cross-sectional and were conducted from 2014 to 2024 across various countries within the SSA region. From all studies 12 (42.86%) were conducted in Ethiopia [37,40,42,46,47,49–51,53–55,62], seven (25.0%) were from Nigeria [7,38,41,48,52,61,63], four (14.29%) from Ghana [43–45,59], and two (7.14%) were from Sudan [56,58] and Kenya [57,60], and one study (3.6%) was conducted in Cameroon [39]. Among those the highest sample size was observed in studies conducted in Sudan [58] which was equal to 1000 and the lowest was found in Nigeria 50 [38]. Out of 28 studies included in this review and analysis, eight (28.57%) were conducted on community-based and 20 (71.43%) were facility-based studies. The articles examined in this study encompass various participant groups, such as pregnant women, women of childbearing age, and postnatal mothers. Out of all the articles, the participants of eight studies (28.57%) were women of childbearing age, 16 studies (57.1%) centered on pregnant mothers, and four studies (14.3%) included postnatal mothers (Table 2). The majority of the studies included in the analysis were of high quality, with quality scores ranging from seven to eight (S2 Table).

### 3.3 Proportion of preconception FA supplementation

The proportion of preconception FA supplementation ranged from 1.9% to 45.2% among the included studies. The highest (45.2%) and lowest (1.9%) proportion of FA supplementation during preconception period were recorded in the southern and Amhara regions of Ethiopia, respectively [47,55]. The pooled proportion using the fixed effect model showed significant heterogeneity among the studies. Therefore, we performed the analyses using the random effects model. Using the random effects model, the estimated pooled proportion of FA supplementation among women during preconception period in SSA was 14.10% (95% CI: 11.22–16.98) with significant heterogeneity between studies ($I^2$ = 97.7%, $p$ = 0.001). The pooled proportion of FA supplementation during preconception period was presented using a forest plot (Fig 2).

### 3.4 Subgroup analysis

A subgroup analysis was done by classifying studies based on corresponding countries in SSA to compute and relate the proportion of preconception FA supplementation focusing on

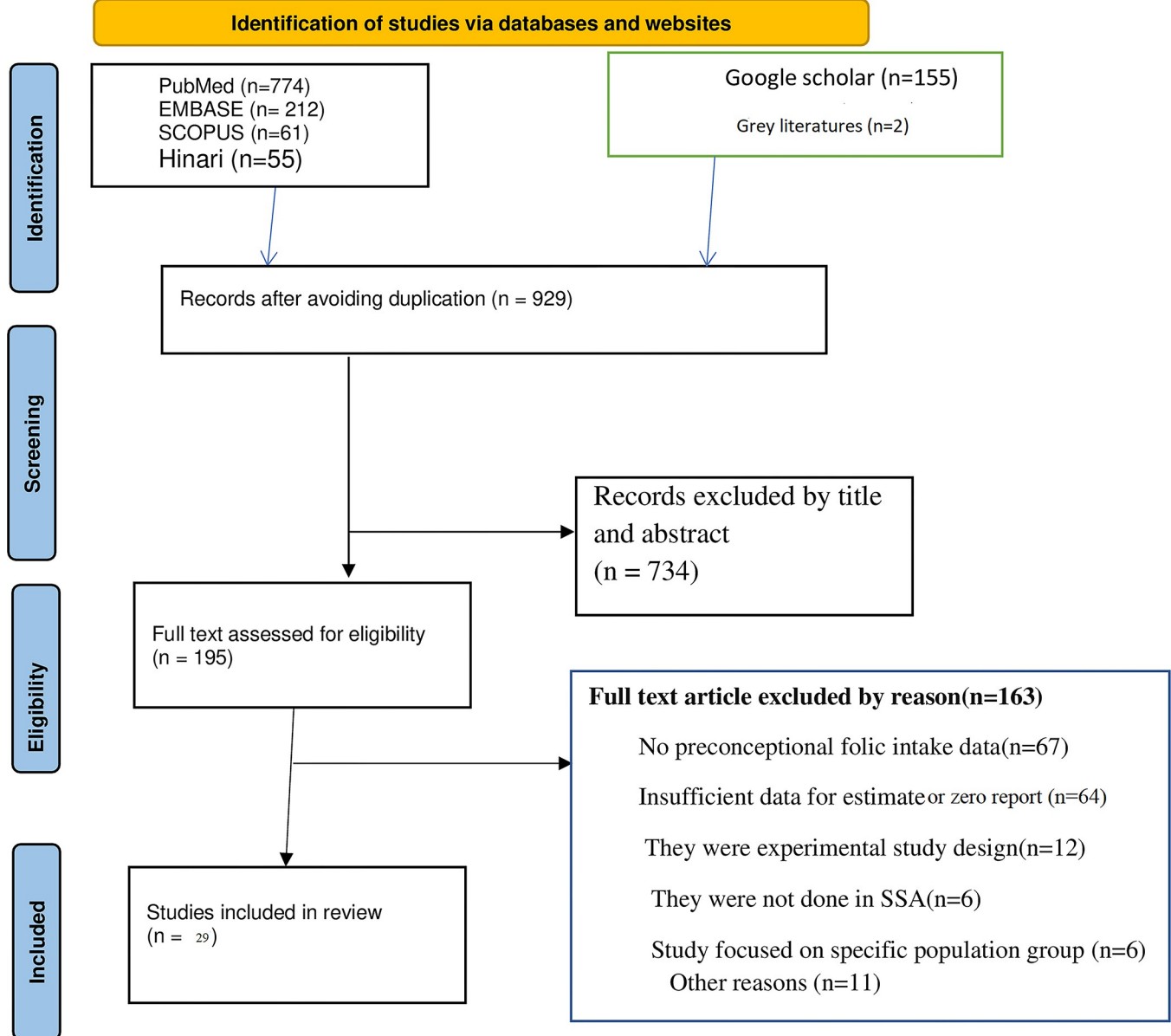

**Fig 1. PRISMA flow diagram of article selection for systematic review and meta-analysis of FA supplementation during preconception period in sub-Saharan African countries.**

various characteristics. Sub-group analysis was done by study location (sub-Africa region and country), study period, and based on types of participants. From the 28 studies, the highest estimated FA supplementation proportion during preconception period was found in studies conducted in Kenya ((22%; 95% CI: 19%-25%), $I^2$ = 97.7%), followed by studies conducted in Ghana ((20%; 95% CI: 7%-33%), $I^2$ = 96.9%) (Fig 3).

The proportion estimates were categorized based on the study period, with studies conducted up to 2019 and those conducted after 2019. However, there was no noticeable difference in proportion estimates between the two study periods (Fig 4).

We examined articles involving different participant categories of mothers including pregnant women, childbearing-age women, and postnatal mothers. Among these groups, the

**Table 2. Characteristics of the included studies and their proportion of preconceptional FA supplementation in SSA, 2024.**

| S. No | Author | Publication year | Study design | Africa region | Country | Study setting | Population | Location/ Scale | Number of Facilities/ scales | Sample size | Reported prevalance (%) |
|---|---|---|---|---|---|---|---|---|---|---|---|
| 1 | Adebo et al. | 2017 | cross sectional | West Africa | Nigeria | institutional based | pregnant women | Facilities | 3 | 300 | 3.0 |
| 2 | Amaje et al. | 2022 | cross sectional | East Africa | Ethiopia | community based | pregnant women | District | 4 | 677 | 12.7 |
| 3 | Anzaku A. | 2014 | cross-sectional | West Africa | Nigeria | institutional based | pregnant women | Facility | 1 | 595 | 4.8 |
| 4 | Asresu et al. | 2019 | cross sectional | East Africa | Ethiopia | community based | delivered mothers | District | 1 | 564 | 15.7 |
| 5 | Ayele A. et al. | 2022 | cross-sectional | East Africa | Ethiopia | community based | childbearing age women | District | 1 | 504 | 8.6 |
| 6 | Boakye Y. et al. | 2018 | cross-sectional | West Africa | Ghana | institutional based | pregnant women | Facility | 1 | 200 | 34.0 |
| 7 | Dessie et al. | 2017 | cross-sectional | East Africa | Ethiopia | institutional based | pregnant women | Facility | 1 | 422 | 1.9 |
| 8 | Ekem et al. | 2018 | cross-sectional | West Africa | Nigeria | institutional based | pregnant women | Facility | 1 | 453 | 3.8 |
| 9 | Fetena N.et al. | 2023 | cross-sectional | East Africa | Ethiopia | community based | pregnant women | District | 1 | 393 | 11.3 |
| 10 | Gedefaw et al. | 2018 | cross-sectional | East Africa | Ethiopia | institutional based | delivered mothers | province | 1 | 222 | 9.0 |
| 11 | Gelgalu et al. | 2021 | cross-sectional | East Africa | Ethiopia | institutional based | pregnant women | Facility | 5 | 455 | 11.0 |
| 12 | Girma et al. | 2023 | cross-sectional | East Africa | Ethiopia | institutional based | pregnant women | Facility | 1 | 385 | 31.2 |
| 13 | Habte et al. | 2021 | cross-sectional | East Africa | Ethiopia | community based | delivered mothers | District | 1 | 600 | 45.2 |
| 14 | Hassan et al. | 2024 | cross-sectional | East Africa | Sudan | institutional based | 1st trimaster Preg | Facility | 1 | 720 | 3.8 |
| 15 | Lawal T. | 2014 | cross-sectional | West Africa | Nigeria | institutional based | childbearing age women | Facility | 2 | 602 | 2.5 |
| 16 | Akinajo et al. | 2019 | Cross sectional | West Africa | Nigeria | institutional based | pregnant women | Facility | 1 | 50 | 32.0 |
| 17 | Alemajo et al. | 2022 | cross sectional | West Africa | Cameroon | institutional based | pregnant women | Facility | 2 | 393 | 5.1 |
| 18 | Alsammani et al. | 2021 | cross sectional | East Africa | Sudan | institutional based | pregnant women | Facility | 1 | 1048 | 3.2 |
| 19 | Asumadu et al. | 2020 | cross-sectional | West Africa | Ghana | institutional based | delivered mothers | Facility | 1 | 363 | 6.9 |
| 20 | Beyuo T et al. | 2021 | cross-sectional | West Africa | Ghana | institutional based | pregnant women | Facility | 1 | 120 | 10.8 |
| 21 | Fekene et al. | 2020 | cross-sectional | East Africa | Ethiopia | community based | childbearing age women | District | 1 | 680 | 7.8 |
| 22 | Gamshe E and DDB | 2021 | cross-sectional | East Africa | Ethiopia | institutional based | pregnant women | Facility | 3 | 333 | 30.2 |
| 23 | Joyce C. et al. | 2018 | cross-sectional | East Africa | Kenya | institutional based | childbearing age women | Facility | 1 | 384 | 19.8 |
| 24 | Mohammed B. et al. | 2019 | cross-sectional | West Africa | Ghana | institutional based | pregnant women | Facility | 18 | 303 | 28.7 |
| 25 | Mukhalisi A et al. | 2022 | cross-sectional | East Africa | Kenya | community based | Couples | District | 1 | 422 | 23.7 |
| 26 | Nwaolisa H. et al | 2021 | cross-sectional | West Africa | Nigeria | institutional based | pregnant women | Facility | 1 | 165 | 38.0 |

*(Continued)*

**Table 2.** (Continued)

| S. No | Author | Publication year | Study design | Africa region | Country | Study setting | Population | Location/ Scale | Number of Facilities/ scales | Sample size | Reported prevalance (%) |
|---|---|---|---|---|---|---|---|---|---|---|---|
| 27 | Olowokere, A.E et. al | 2015 | cross-sectional | West Africa | Nigeria | institutional based | childbearing age women | Facility | 11 | 375 | 30.4 |
| 28 | Setegn Alie M et al. | 2022 | cross-sectional | East Africa | Ethiopia | community based | childbearing age women | District | 1 | 624 | 5.3 |
| 29 | Ubong Akpan Okon et al. | 2020 | cross-sectional | West Africa | Nigeria | institutional based | childbearing age women | Facility | 11 | 606 | 10.4 |

highest estimated proportion of FA supplementation was observed among studies with postnatal mothers as participants (19.2%; 95% CI: 3.8%, 34.5%) with level of heterogeneity ($I^2 =$ 98.9%) (Fig 5).

Subgroup analyses indicated that the proportion of FA supplementation before pregnancy showed similar magnitude variations across regions implying a consistent pattern across the sub-regions of Africa (Fig 6).

### 3.5 Heterogeneity and sensitivity analysis

In our systematic review and meta-analysis, we addressed heterogeneity in study results by initially using a fixed-effects model and subsequently by using a random-effects model to accommodate variability. Despite conducting sensitivity analyses, subgroup analyses, and meta-regression analyses, significant heterogeneity persisted, leading to further investigation. Upon conducting a meta-regression analysis, we found that the year of publication did not influence heterogeneity. However, we discovered that the sample size had a statistically significant relation to the presence of heterogeneity, as shown in Table 3.

Sensitivity analysis was performed to assess the impact of each individual study on the overall proportion estimate. Remarkably, removing any single study did not result in a statistically significant change to the pooled proportion estimate. Additionally, we conducted sensitivity analyses excluding studies of poor quality. Even after excluding these lower-quality studies, there were no significant changes to the estimated proportion (Fig 7).

### 3.6 Publication bias

In our meta-analysis, evidence of publication bias was observed through funnel plots and tests, indicating asymmetrical distribution (Fig 8).

Trim and fill analysis was conducted to estimate the effect sizes of missed studies and adjust for publication bias (Fig 9).

Egger's and Begg's tests were performed, showing considerable evidence of publication bias (Table 4).

### 3.7 Evidence certainty

Overall, the certainty of the evidence regarding the pooled proportion estimates assessed by the GRADE approach was low (Table 5). Although the majority of the included studies were of high quality, significant heterogeneity was observed, with an $I^2$ value exceeding 97.7%. The directness of the evidence was rated as direct, and the precision of the proportion estimate was satisfactory. However, evidence of publication bias contributed to downgrading the certainty. Consequently, while the evidence offers valuable insights into the proportion of preconception

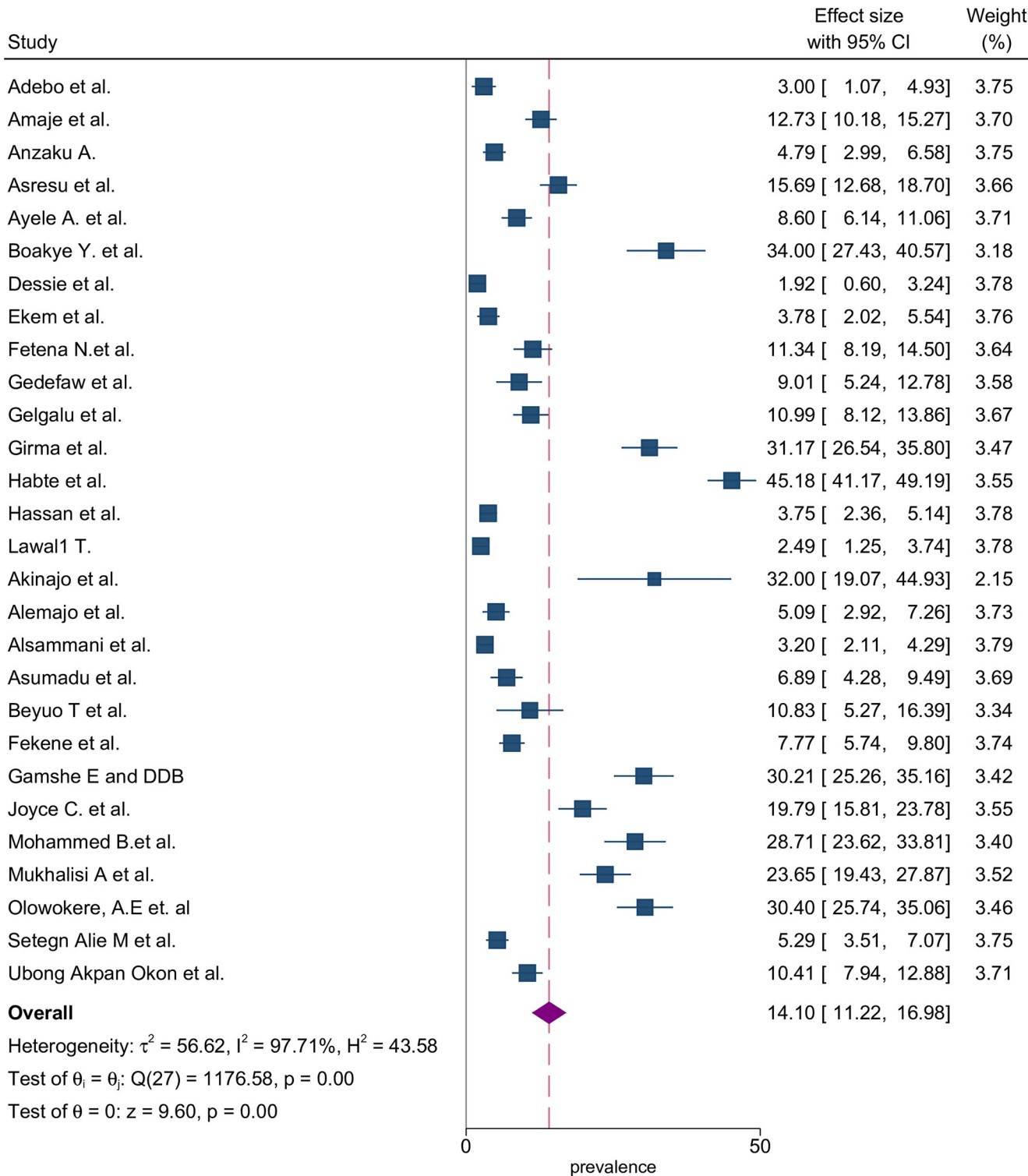

**Fig 2. Forest plot of FA supplementation during preconception period in sub-Saharan African countries.**

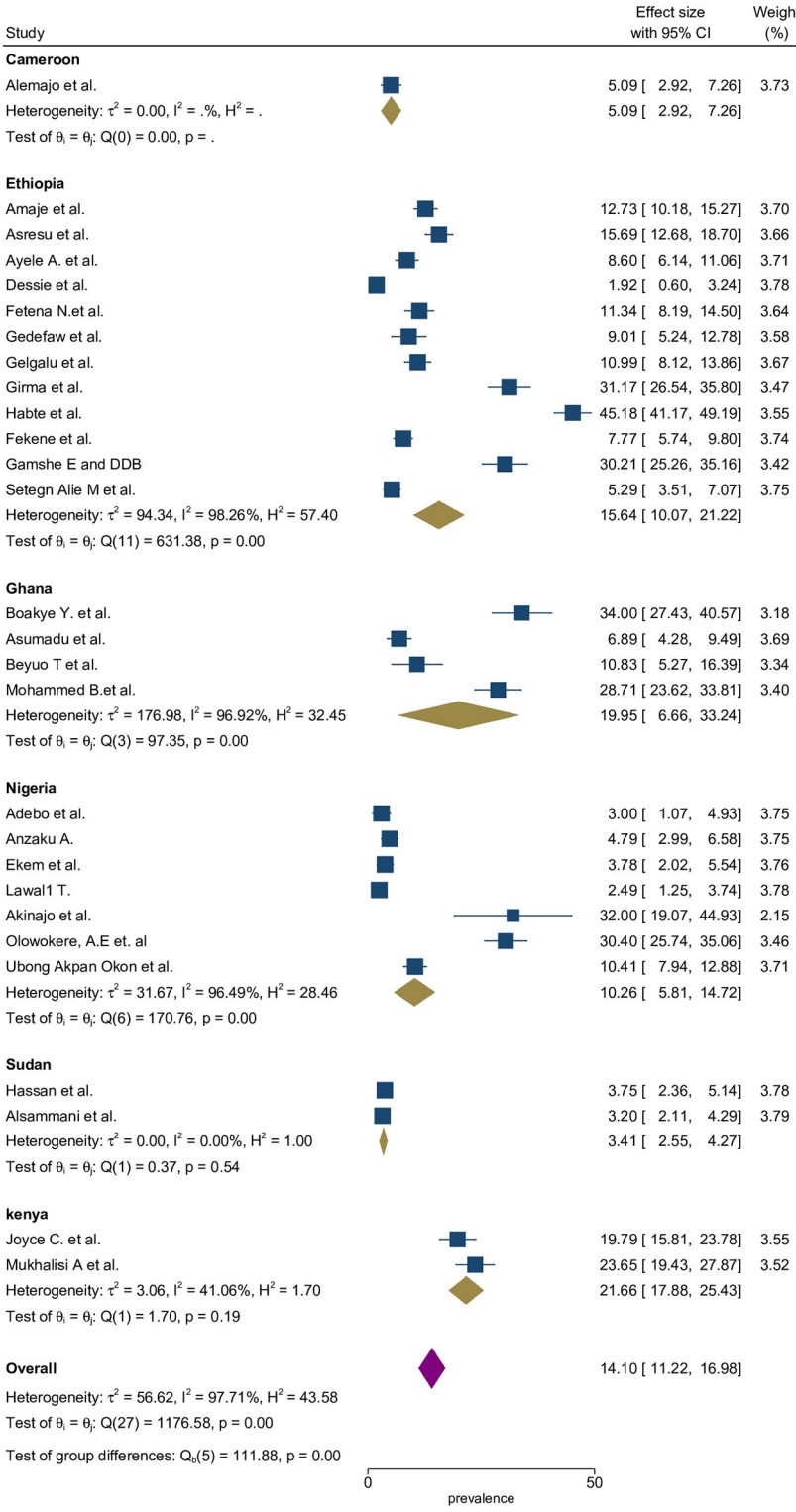

**Fig 3. Sub-group by country pooled proportion of FA supplementation during preconception period in SSA countries.**

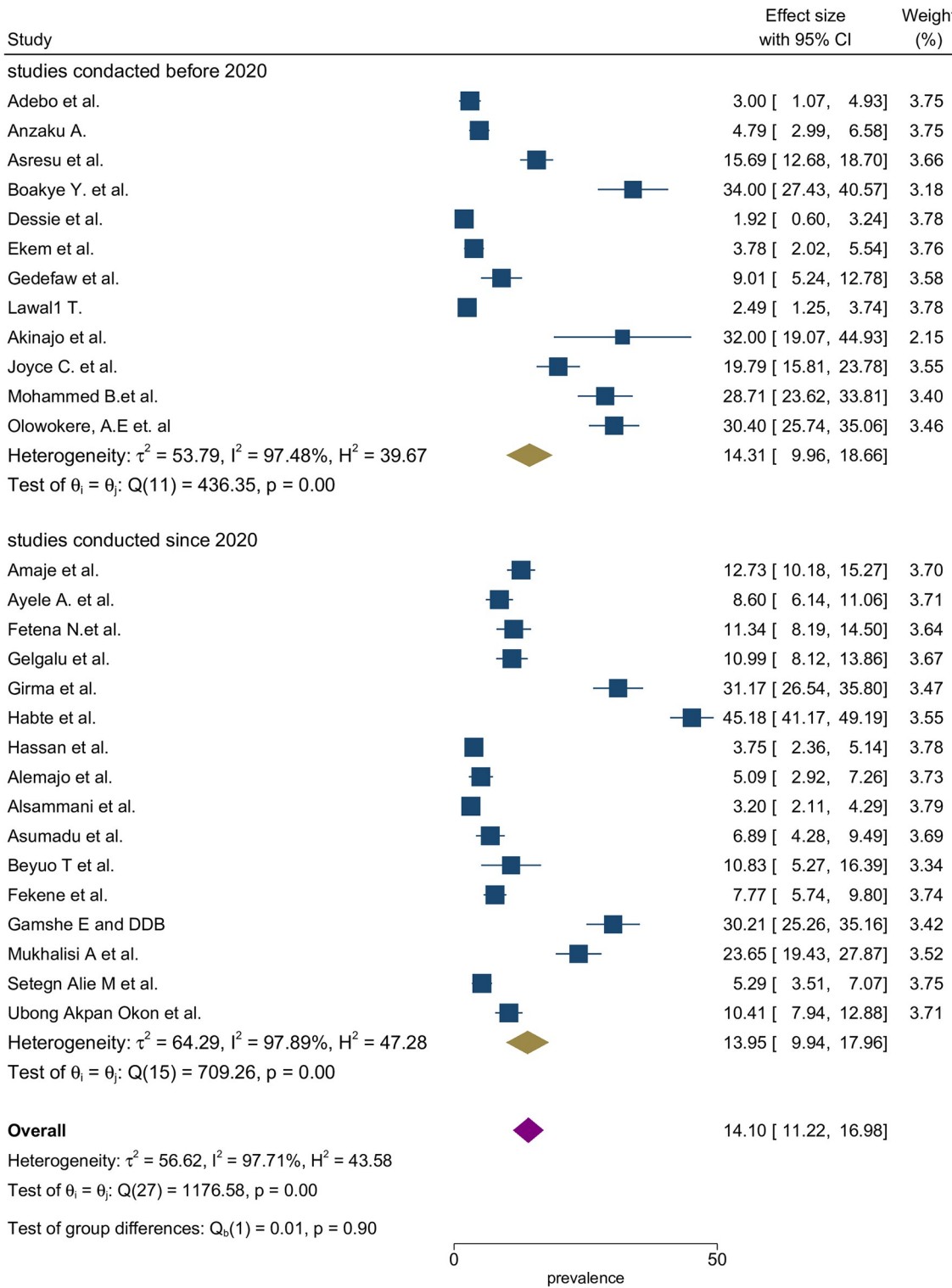

**Fig 4. Sub-group by year pooled proportion of folic acid supplementation during preconception period in SSA countries.**

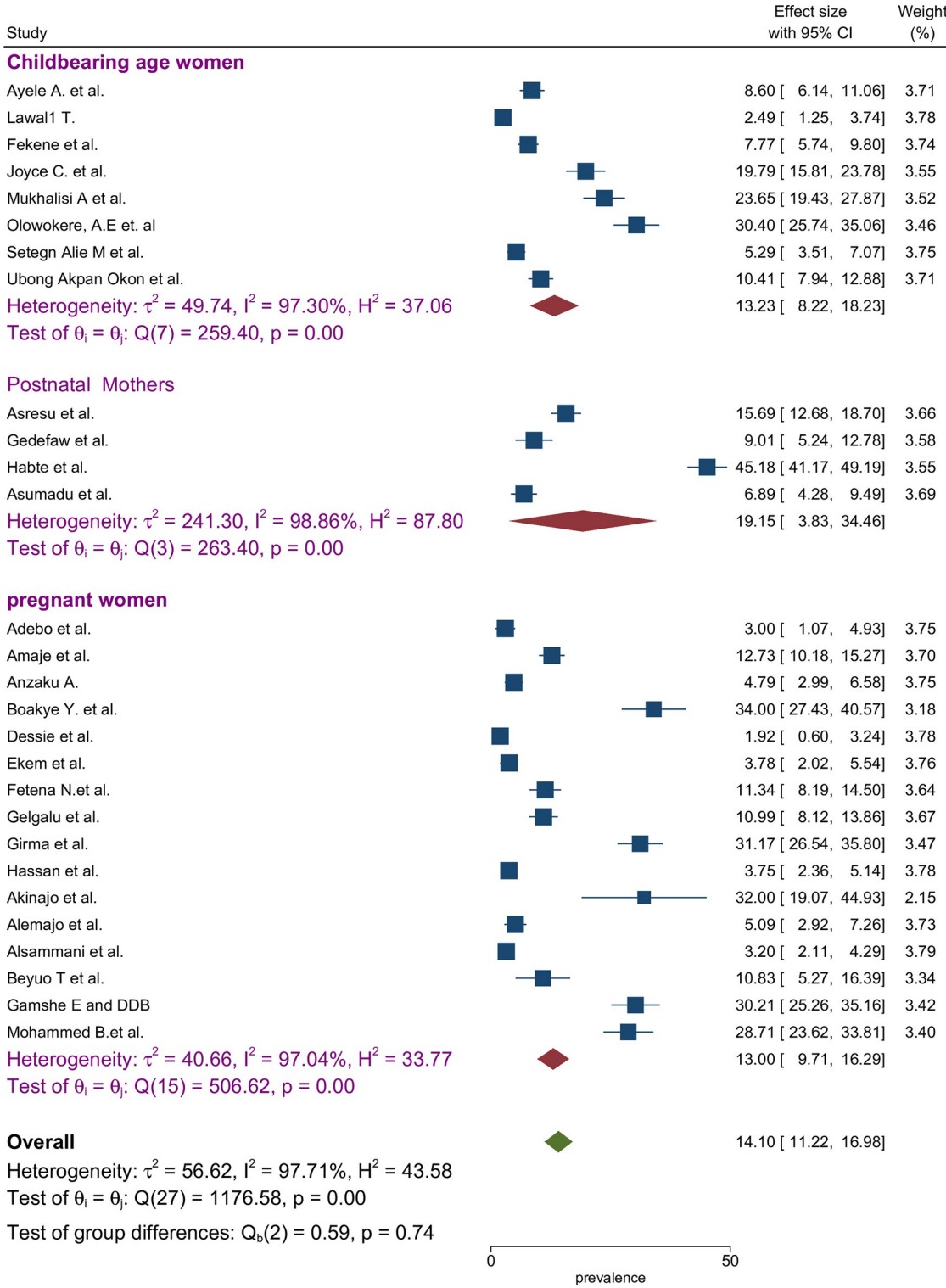

**Fig 5. Sub-group by participant categories pooled proportion of FA supplementation during preconception period in SSA countries.**

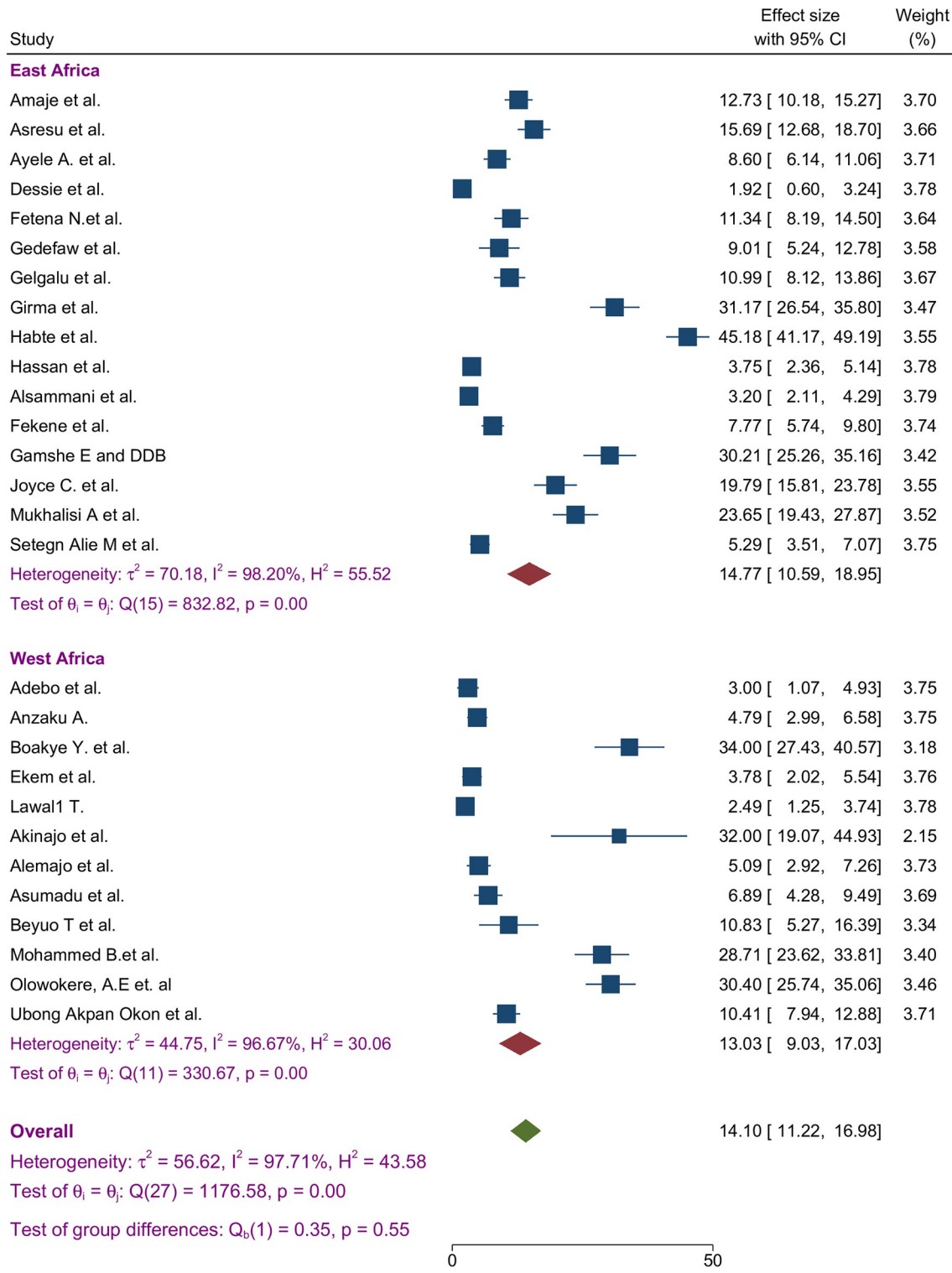

**Fig 6. Sub-group by African region pooled proportion of FA supplementation during preconception period in SSA countries.**

**Table 3. Meta-regression analysis of factors with heterogeneity of the proportion of preconceptional FA supplementation in SSA, 2024.**

| Heterogeneity source | Coefficient | Std. err. | P-value |
|---|---|---|---|
| Sample size | -.023446 | .0080509 | 0.004 |
| Publication year | .7000846 | .5774253 | 0.225 |
| _cons | -1389.062 | 1165.572 | |

folic acid supplementation among women in Sub-Saharan Africa, it is essential to recognize the limitations and uncertainties inherent in the findings.

## 4. Discussion

The current systematic review provides evidence of an estimated pooled proportion of FA supplementation during the preconception period in SSA countries. The study synthesizes data from 28 cross-sectional studies conducted between 2014 and 2024, involving 12,562 participants, with significant contributions from Ethiopia (42.9%) and Nigeria (25%). The proportion of preconception FA supplementation reported in included studies varied widely, ranging from 1.9% to 45.2%. In the meta-analysis utilizing a random-effects model, the subgroup and

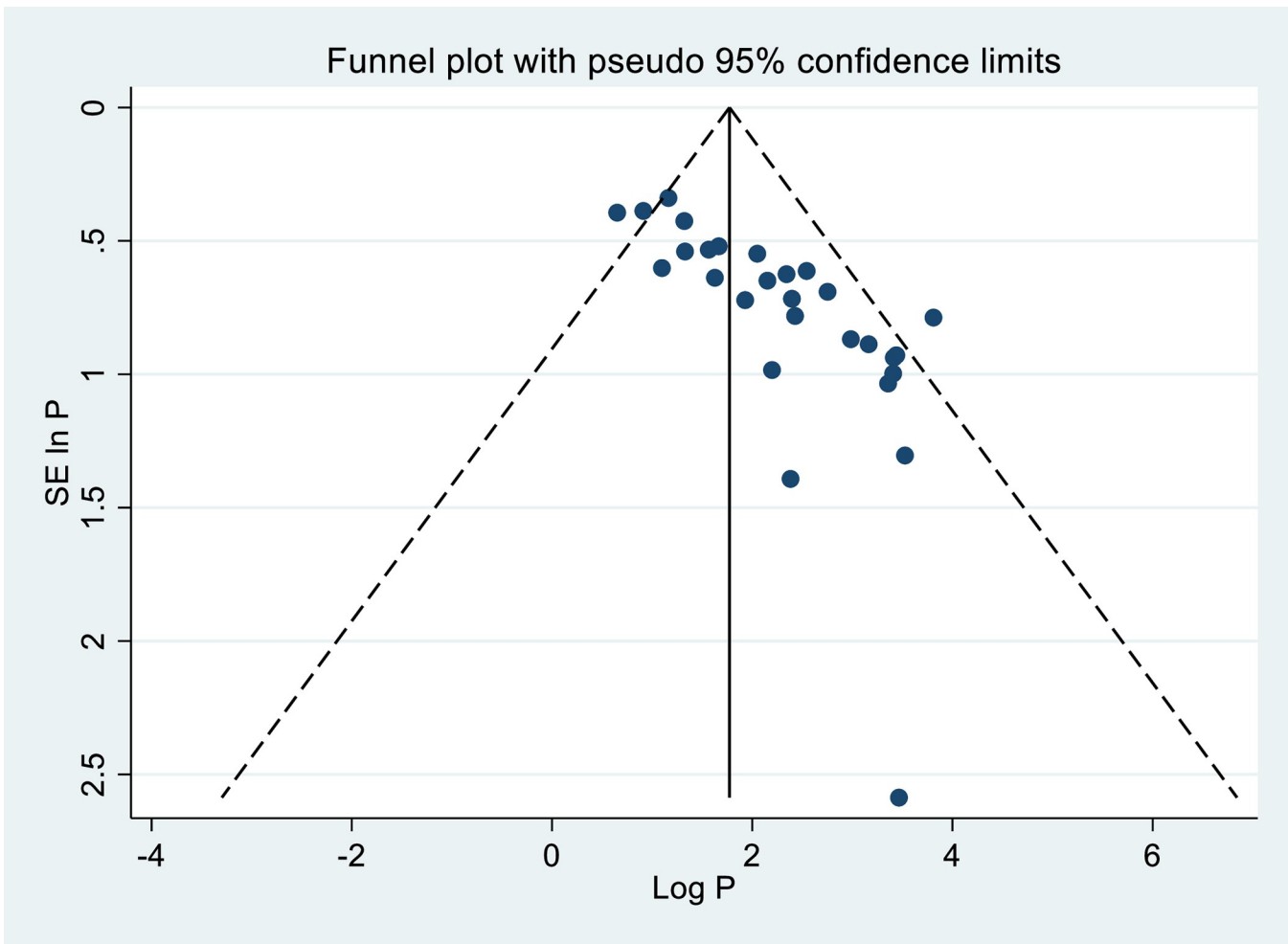

**Fig 7. Sensitivity** *analysis* **of FA** *supplementation during preconception period* **in SSA** *countries.*

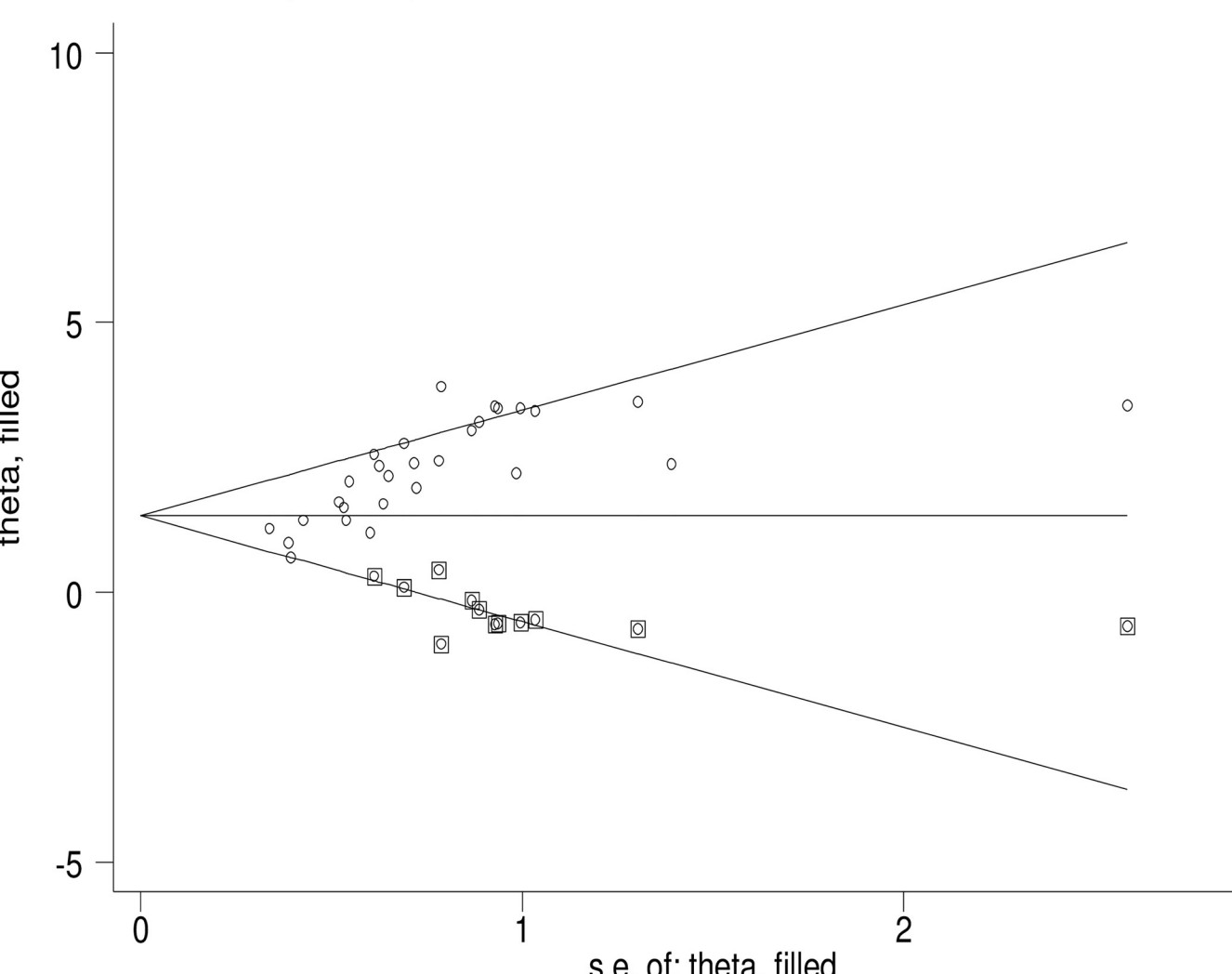

**Fig 8. Funnel plot result of FA supplementation during preconception period in SSA countries.**

meta-regression analysis. Additionally, the quality assessment of the included studies showed that the majority were of high quality, demonstrating robust methodologies and reliable data.

The quality of evidence for preconception folic acid supplementation, evaluated with the GRADE approach, was rated as low. This is primarily due to notable differences between studies and some concerns about publication bias, despite the fact that most of the included studies were of high quality. Subgroup analysis by African region reveals closely similar findings between Eastern and Western African countries.

Despite implementing various strategies such as targeted health education initiatives, policy reforms, and healthcare provider training programs, the subgroup analysis found no significant change in the proportion of FA supplementation across different study periods. This implies a concerning trend indicating a persistent lack of improvement in FA utilization and failure to align with WHO recommendations which suggests a critical need for reevaluation and potential enhancement of current intervention approaches to effectively address barriers hindering optimal FA before pregnancy across SSA countries. This meta-analysis showed that

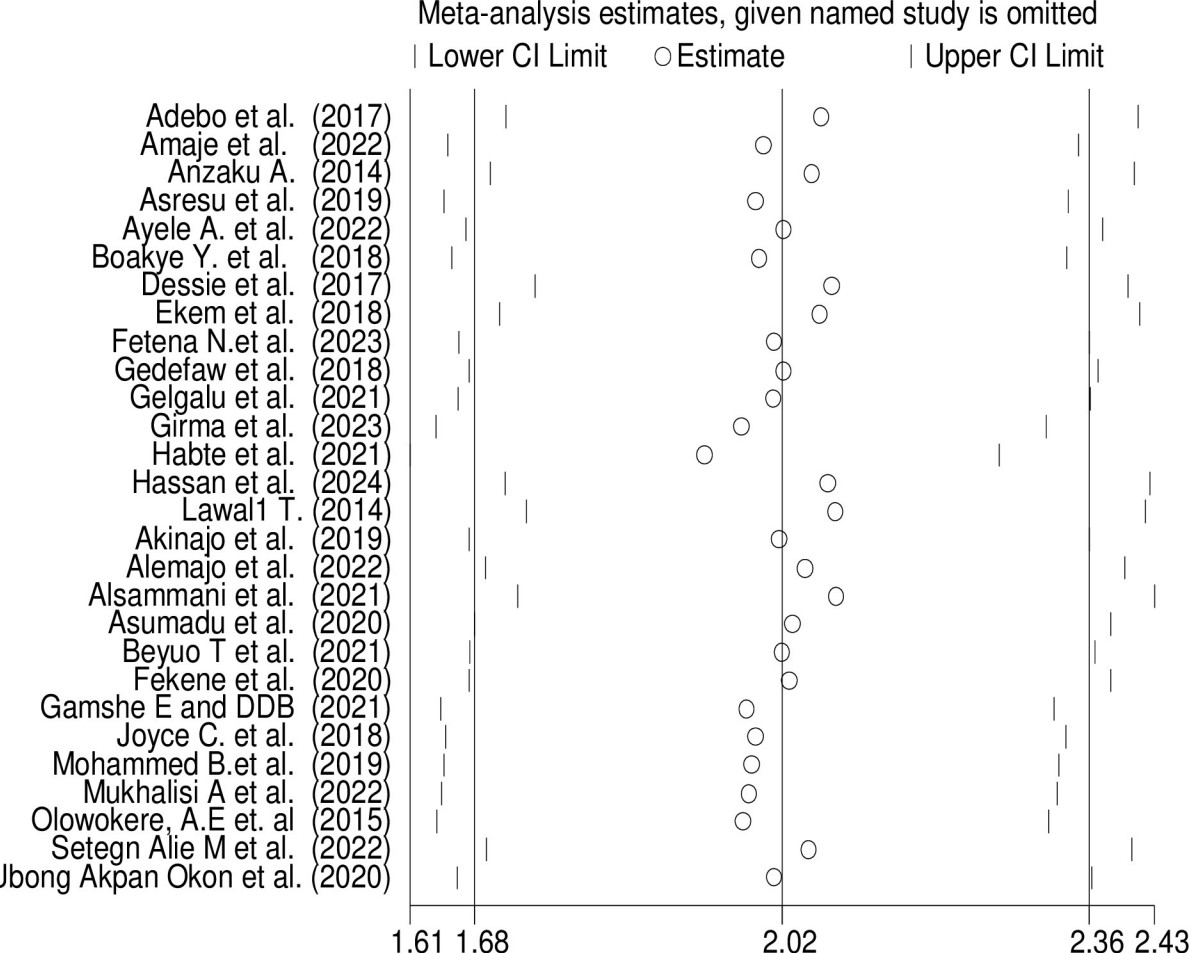

**Fig 9. Trim and fill analysis of FA supplementation during preconception period in SSA countries.**

the highest estimated proportion FA supplementation during preconception period was found in studies conducted in Kenya (22%). Another reason for this variation might be due to the small number of articles from Kenya and these small studies may overestimate the true effect size of FA supplementation during the preconception period. It is plausible that the observed variability in FA supplementation during preconception period rates across countries could be linked to the educational levels, cultural norms, and different activities and efforts to increase the supplementation of FA before conception.

In subgroup analysis based on study participants, the highest estimated proportion of FA supplementation was observed among studies involving postnatal mothers (19.2%). The observed variation in FA supplementation proportion during preconception period may be attributed to the positive influence of postnatal care (PNC) services. Mothers who receive postnatal services have higher health-seeking behavior for their health and their newborns [65].

**Table 4. Egger's of publication bias of included studies in the systematic review and meta-analysis of preconceptional FA supplementation in SSA, 2024.**

| Std_Eff | Coefficient | Std. err. | t | P>\|t | [95% conf. | interval] |
|---|---|---|---|---|---|---|
| slope | .0443915 | .2695872 | 0.16 | 0.870 | -5097529 | .5985359 |
| bias | 2.98436 | .4336036 | 6.88 | 0.000 | 2.093075 | 3.875644 |

**Table 5. Certainty assessment for included outcomes for the proportion of preconception FA supplementation among women in Sub-Saharan Africa.**

| Outcome | Study design | Source | Risk of bias | Inconsistency | Directness | Imprecision | Publication bias | Proportion (CI) | Certainty |
|---|---|---|---|---|---|---|---|---|---|
| Preconception folic acid supplementation | Observational studies | 28 | Low[1] | Serious[2] | Direct[3] | low[4] | Very low[5], | 14.10% (11.2216.98) | Low |

[1]Rated as low because most studies included were of high quality.

[2]Rated as Serious due to significant heterogeneity ($I^2 > 97.7\%$).

[3] Rated as direct as there is a clear link between preconception folic acid intake and the outcome.

[4]Rated as low due to narrow confidence intervals.

[5] publication bias present and rated as low but handled with trim and fill analysis.

This increased awareness and active engagement with healthcare services potentially plays a pivotal role in fostering improved adherence to preconception care practices, including the essential supplementation of FA.

Our findings demonstrate a higher proportion of preconceptional FA supplementation compared to a study conducted in Egypt, where only 8.8% of women reported taking FA supplements before pregnancy [66]. The discrepancy in proportion might be attributed to the time period of the conducted studies. In recent years, various mechanisms have been implemented, especially in low-income countries, to enhance awareness and access to FA supplementation. Many studies included in this meta-analysis were conducted after 2014, reflecting these advancements, which likely contributed to an increased proportion of preconceptional FA use observed in more recent studies compared to those conducted earlier. Another reason for this discrepancy could stem from variations in research methodology and the distinct characteristics of the population groups studied.

In addition, Toivonen et al. showed that the proportion of FA supplementation during the preconception period in Africa was 0% [28], which is lower than from our estimated pooled proportion. The discrepancy might be due to the previous review including only three studies from the African region, potentially underestimating the true proportion of FA supplementation during the preconception period. Another contributing factor to the variation in proportion estimates could be attributed to several factors, including changes in healthcare policies, economic conditions, and awareness campaigns.

The findings of this study indicate a significantly lower proportion of preconceptional FA supplementation compared to a study conducted in Lebanon, where 48% of participants reported using FA supplements before pregnancy [67]. This discrepancy underscores variations in health behaviors and practices across different populations, influenced by factors such as healthcare access, awareness campaigns, socioeconomic status, and cultural norms related to maternal health.

Similarly, our findings showed a significantly lower proportion compared to a study conducted in Vietnam, where FA use before pregnancy was reported to be 24.5% [68]. The lower proportion of FA supplementation in SSA compared to Vietnam can be attributed to factors such as less strong public health initiatives, socioeconomic disparities, and varying cultural practices that health seeking behavior. In the prior study, a significant majority of participants, exceeding 77%, had planned pregnancies which drove them to actively seek preconception care services including FA supplementation [69,70].

Likewise, studies conducted across various high-income nations have indicated a higher proportion of preconceptional FA intake compared to the current findings. For instance, a systematic review and meta-analysis conducted in 2018 reported that in North America, the proportion of preconceptional FA supplementation ranged from 32% to 51% [28]. Similarly,

research conducted in China, Iran, and Vietnam documented proportion rates of 52.1%, 54.5%, and 24.9%, respectively [68,71,72]. These differences in supplementation rates across regions can be attributed to various factors, such as disparities in healthcare policies, availability of supplements, and the effectiveness of public awareness campaigns.

In high-income countries, comprehensive health education and awareness programs for childbearing women are provided through well-resourced healthcare systems, public health campaigns, and mandatory prenatal care guidelines. Greater awareness and education about the importance of preconceptional health and nutrition in high-income countries contribute significantly to the higher proportion of FA supplementation [73]. In contrast, low-income nations face significant barriers such as financial constraints and limited access to healthcare facilities, which pose challenges to women seeking essential healthcare services [74].

The findings highlight a critical need for targeted public health strategies aimed at improving access, awareness, and utilization of FA supplementation across Sub-Saharan Africa. In conclusion, the findings provide valuable insights into the proportion of preconception folic acid supplementation among women in Sub-Saharan Africa. However, based on the GRADE framework, the overall quality of evidence is rated as low. These limitations underscore the need for caution in interpreting the results and highlight the necessity for further research to improve the reliability of findings in this important area.

## 5. Strengths and limitations of the review

This systematic review and meta-analysis is the first comprehensive assessment of preconceptional FA supplementation in SSA, highlighting several strengths. This review is notably strong in its broad scope, covering a wide range of studies from different countries and regions within SSA. It includes both published and unpublished articles in its systematic review and meta-analysis across the continent. Subgroup analysis was performed to minimize statistical heterogeneity. While our study highlights the state of FA supplementation during preconception period in SSA, it is important to acknowledge several limitations. The lack of studies representing all countries across SSA may restrict its ability to be generalized to the entire region. Significant heterogeneity and potential publication bias require cautious interpretation. Additionally, comparing findings is challenging due to a lack of regional and global systematic reviews.

## 6. Conclusions and recommendations

The systematic review and metaanalysis revealed alarmingly low rates of FA supplementation during the preconception period in SSA. Addressing this critical issue necessitates coordinated efforts from policymakers, healthcare providers, and community stakeholders. Key strategies should focus on increasing awareness among women of reproductive age and healthcare professionals about the benefits of FA supplementation. Furthermore, integrating FA supplementation into existing maternal and child health programs, strengthening partnerships with pharmaceutical companies, and advocating for policy reforms to ensure affordable access to supplements are crucial steps toward improving maternal and child health outcomes in the region.

Future research should prioritize longitudinal studies and qualitative investigations into cultural factors, alongside assessing targeted interventions. These efforts are essential for effectively addressing disparities and enhancing maternal and child health outcomes in the region. Updated proportion estimates of FA supplementation before pregnancy are crucial for informing policymakers, health planners, communities, as well as governmental and non-governmental organizations.

## Supporting information

**S1 Table. PRISMA checklist.**
(DOCX)

**S2 Table. Keywords and searching methodology.**
(DOCX)

**S3 Table. Methodological quality assessment of included studies using Newcastle-Ottawa Scale (NOS) for preconceptional FA supplementation in SSA, 2024.**
(DOCX)

**S1 Appendix. Individual studies with exclusion.**
(DOCX)

**S2 Appendix. Data extraction form with data extractors name and date.**
(XLSX)

## Acknowledgments

The authors would like to thank all authors of studies included in this systematic review and meta-analysis. Additionally, we extend our gratitude to Aklilu Endalamaw for his valuable support throughout the study.

## Author Contributions

**Conceptualization:** Mekuriaw Nibret Aweke, Aysheshim Kassahun Belew, Esmael Ali Muhammad, Berhanu Mengistu.

**Data curation:** Lemlem Daniel Baffa, Ever Siyoum Shewarega, Aysheshim Kassahun Belew.

**Formal analysis:** Mekuriaw Nibret Aweke, Lemlem Daniel Baffa, Ever Siyoum Shewarega, Aysheshim Kassahun Belew, Esmael Ali Muhammad, Berhanu Mengistu.

**Investigation:** Mekuriaw Nibret Aweke.

**Methodology:** Mekuriaw Nibret Aweke, Elsa Awoke Fentie, Muluken Chanie Agimas, Lemlem Daniel Baffa, Ever Siyoum Shewarega, Aysheshim Kassahun Belew, Berhanu Mengistu.

**Resources:** Mekuriaw Nibret Aweke.

**Software:** Mekuriaw Nibret Aweke, Elsa Awoke Fentie, Muluken Chanie Agimas, Ever Siyoum Shewarega, Aysheshim Kassahun Belew, Esmael Ali Muhammad.

**Visualization:** Mekuriaw Nibret Aweke, Lemlem Daniel Baffa, Berhanu Mengistu.

**Writing – original draft:** Mekuriaw Nibret Aweke, Muluken Chanie Agimas, Lemlem Daniel Baffa, Ever Siyoum Shewarega, Aysheshim Kassahun Belew.

**Writing – review & editing:** Mekuriaw Nibret Aweke, Esmael Ali Muhammad, Berhanu Mengistu.

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
