## [Decision Letter · Decision Letter 0]

1 Jul 2024

PONE-D-24-13717Folic acid supplementation during preconception period in sub-Saharan African countries: A systematic review and meta-analysisPLOS ONE

Dear Dr. Aweke,

Thank you for submitting your manuscript to PLOS ONE. After careful consideration, we feel that it has merit but does not fully meet PLOS ONE’s publication criteria as it currently stands. Therefore, we invite you to submit a revised version of the manuscript that addresses the points raised during the review process.

We look forward to receiving your revised manuscript.

Kind regards,

Jai K Das

Academic Editor

PLOS ONE

Journal Requirements:

https://www.iapsmupuk.org/journal/index.php/IJCH/article/view/2368

https://link.springer.com/article/10.1007/s00381-023-05910-7?

https://link.springer.com/article/10.1007/s00381-023-05910-7?

In your revision ensure you cite all your sources (including your own works), and quote or rephrase any duplicated text outside the methods section. Further consideration is dependent on these concerns being addressed.

"The authors declare that they have no conflicts of interest."

4. We note that you have referenced (unpublished) on page 6, which has currently not yet been accepted for publication. Please remove this from your References and amend this to state in the body of your manuscript: (ie “Bewick et al. [Unpublished]”) as detailed online in our guide for authors

Reviewers' comments:

Reviewer's Responses to Questions

**Comments to the Author**

1. Is the manuscript technically sound, and do the data support the conclusions?

Reviewer #1: Yes

Reviewer #2: Partly

2. Has the statistical analysis been performed appropriately and rigorously? 

Reviewer #1: Yes

Reviewer #2: Yes

3. Have the authors made all data underlying the findings in their manuscript fully available?

Reviewer #1: Yes

Reviewer #2: No

4. Is the manuscript presented in an intelligible fashion and written in standard English?

Reviewer #1: No

Reviewer #2: No

5. Review Comments to the Author

Reviewer #1: The manuscript under review addresses a critical question of folic acid supplementation in sub-sharan Africa. The methodolgy, analysis and results are relevant, however, the paper requires careful reconsideration and possibly re-writing the introduction and the discussion sections to strengthen the manuscript. Some specific points to consider are highlighted below:

Abstract:

1. Add the number of studies and participants included in the review in the results section of the abstract.

Introduction:

1. NTDs has been spelled out more than once. Spell out once when used for the first time and then use the acronym for the rest of the manuscript.

2. Some of the references in the introduction are very old. I suggest replacing these with the more recent references.

3. I found the introduction very weak. There are quite a few recent relevant references assessing the existing evidence, guidelines, and policies around folic acid supplementation. I suggest that the authors look at these papers and add some of the relevant references in their introduction to make a stronger case for the review:

- Quinn M, Halsey J, Sherliker P, Pan H, Chen Z, Bennett DA, Clarke R. Global heterogeneity in folic acid fortification policies and implications for prevention of neural tube defects and stroke: a systematic review. EClinicalMedicine. 2024 Jan 1;67.

- Viswanathan M, Urrutia RP, Hudson KN, Middleton JC, Kahwati LC. Folic acid supplementation to prevent neural tube defects: a limited systematic review update for the US preventive services task force.

- Barry MJ, Nicholson WK, Silverstein M, Chelmow D, Coker TR, Davis EM, Donahue KE, Jaén CR, Li L, Ogedegbe G, Rao G. Folic acid supplementation to prevent neural tube defects: US Preventive Services Task Force reaffirmation recommendation statement. Jama. 2023 Aug 1;330(5):454-9.

- De‐Regil LM, Peña‐Rosas JP, Fernández‐Gaxiola AC, Rayco‐Solon P. Effects and safety of periconceptional oral folate supplementation for preventing birth defects. Cochrane database of systematic reviews. 2015(12).

4. Add some references (if available) on country specific prevalance rates of NTDs in Africa.

5. The acronym for sub-sahraan Africa (SSA), folic acid (FA) and low- middle- income countries (LMICs) have been used inconsistently throughout the manuscripot. I would suggest that the authors spell it out when used for first time in the darft and then consistently use the acronym throughout.

Methods:

1. Not sure what this sentence under 'study selection' means: "articles examining the prevalence and determinants of folic acid supplementation during preconception period was gathered from various". From the objectives and results, it seems that only prevalance was assessed. I am not sure whether the authors ahve also looked at the determinants? and if they have, this has not be reported anywhere in the draft.

2. Not sure what this sentence means: "A quantitative cross-sectional study design was employed to enhance clarity and significance." Do you mean that the eligibility for study selection was cross-sectional study design? If yes, then state this clearly.

3. Suggest authors add a table or a figure depicting the regions and the respective countries included in sub-saharan Africa. It would add clarity for the readers as teh regions and respective countries have been disucssed and analysed in the resultss and discussion sections.

4. Studydesign is stated twice under data extraction.

Results:

1. I cant see the full characterictis of the inlcuded studies table (probably because it was in a landscape mode) so I am not sure if my comments are relevant. But I would suggest that the authors delete the columns on study period, center, sampling method, and response rate. I would also suggest that the authors add the columns on sample size and study results (reported prevalance) in the study characteristics table.

2. I would suggets that the authors remove the study quality column from the study characteristics table and add a separate section on study quality.

Discussion:

I would suggest that the authors re-organise the discussion section starting with summarising the reveiw findings, quality of the included studies, comparing the review findings with other studies/reviews, strengths and limitations of the review and research and practice implications.

I would suggest that the authors compare the study findings with studies conducted in similar context. For eaxmple, reference 59 is a study from Italy and hence widely differs in context. However, authors can broadly discuss the differences in the folic acid supplementation prevalance between high income and low- middle- income countries.

Reviewer #2: Thank you for giving me the opportunity to review your paper. I think you paper requires thorough read. Methodology is not clear, few sentances are repetitive, there are grammatical errors and information looks scattered.

Abstract:

Introduction:

Seems very long for an abstract.

Just a suggestion: instead of reporting global estimates of NTDs I would suggest writing the estimates from Sub Saharan Africa.

I dont think daily dosage is required here.

Methods:

Please add on what type of studies did you include? Last date of search? if GRADE was conducted and how was quality of included studies conducted (if you did any of these)

Results:

Line 44: not sure what do you mean by 'individual studies' here.

Line 46: subgroup analysis based on?

Line 49: remove double brackets from the end

Quality of included studies in missing

Introduction

If you have introduced the abbreviation once, then you dont have to introduce it again and again. e.g., NTDs.

Methods:

I would suggest to rearrange some headings. After registration, you should talk about the eligibility criteria and then search strategy and study selection and so on.

Line 117: you have stated that you search institutional repositories. Please state which were those?

Line 120: Please check punctuation: "supplementation, and folic acid administration. folic acid adherence, folinic acid, and vitamin B9."

Line 127-129: This sentence is very vague "A quantitative cross-sectional study design was employed to enhance clarity and significance'

Line 127: 'omitted' is not an appropraite word to use here

I think you need to revisit this paragraph. It requires language editing

Linme 127-129: " A quantitative cross-sectional study design was employed to enhance clarity and significance. In contrast, studies utilizing qualitative methodologies were omitted due to the specific nature of the review and the analytical approach selected for this review" I think this should be discussed under section 2.4.

Line 131: I understand you used RAYYAN then why did you exported data on Endnote?

Section 2.3 says study selection but it doesnt talks clearly about it. You have not written about title/abstract and full text screening. How conflicts were managed? Lacks lot of important information.

Section 2.4: A lot of important information is missing. I am confused. Did you include cross sectional studies and grey literature or only cross sectional studies.

Line 136: better to use the word 'included' rather than incorporated.

Suggest adding a Table on PECO criteria

Line 146: suggest using the word 'resolved' instead of 'fixed'

Line 153 and 155: please avoild using such vague words/phrases: 'the required information' 'important parameters'

Line 157: study setting in repeated twice

Line 156: If you are only including cross-sectional studies then why are you collecting data on study designs.

Line 162: Add reference of Stata

Line 166-168: The sentances are repetitive.

A lot of missing information in methods sections: did you conduct cross-referencing. did you felt need of contacting authors for missing data. You stated that you searched institutional website there is no mention of inclusion of reports in the eligibility criteria.

Why did you not assess the quality of evidence using GRADE?

On what basis did you conduct the sensitivity analysis? high heterogeneity or low quality of studies? How did you "assess the influence of individual studies on pooled estimates"?

Results:

Please Format your headings: "3.1 selection and Identification of studies"

Why did you exclud study based on poor quality? You could have done a sensitivity analysis for it

Line 198: spell out 2. Suggest spelling out numbers <10

Please dont use the word 'individual studies'.. you can say 'included studies'

Table 1 doesn't fit the page and is not clear. what does NS mean? what is R. rate? response rate?

Not sure why did you do subgroup analysis based on study period? What would you get from it?

Quality of included studies MISSING! in text.

Discussion:

I would suggest that you should summarise your study findings in the first para and then discuss about other studies

Lines 270-274: Can be moved later in the discussion under strengths.

Lines 289-291: not sure why are you comparing prevalence of Italy with your results? There is no link. These are two different regions/countries. You can compare with LMICs buy why with Italy?

You have not discussed on how quality of studies can affect the findings of your study?

Conclusion would come after limitations and strengths

Figure 1:

Why were studies excluded based on title and abstract twice?

General Comments:

Did you find studies on peri-conception? if yes, then I would suggest doing a subgroup analysis based on preconception and periconception supplementation.

Why did you included studies on pregnant women when you intention was preconception? Did they collect reterospective data?

What were the ages of the included participants? suggest doing a subgroup analysis bases of age and see the prevalence in young people

Did you do sensitivity analysis based on low quality of studies? if not then suggest doing that too.

6. PLOS authors have the option to publish the peer review history of their article (what does this mean?). If published, this will include your full peer review and any attached files.

Reviewer #1: No

Reviewer #2: No

---

## [Author Response · Author response to Decision Letter 0]

20 Jul 2024

Date: July 19, 2024

To: PLOS ONE, Editorial Office

Subject: Submitting a Revised Version of Manuscript and Point-by-Point Response

Manuscript ID: PONE-D-24-13717

Title: Folic Acid Supplementation During Preconception Period in Sub-Saharan African Countries: A Systematic Review and Meta-Analysis

Dear Editorial Office,

We are pleased to submit the revised version of our manuscript titled "Folic Acid Supplementation During Preconception Period in Sub-Saharan African Countries: A Systematic Review and Meta-Analysis" (Manuscript ID: PONE-D-24-13717). In response to the academic editor and reviewers' comments and suggestions, we have made substantial revisions to improve the clarity and quality of our manuscript. We believe these changes have strengthened our study, and we are grateful for the constructive feedback provided.

Enclosed with this letter, please find our detailed point-by-point response to the reviewers' comments. Each comment has been addressed meticulously, and the corresponding revisions are highlighted in the manuscript for your convenience.

We appreciate the opportunity to revise and resubmit our work to PLOS ONE. We look forward to your positive response.

Regards,

Authors

Point-by-Point Response to the Academic Editor and Reviewers' Comments

1. Comments from the Academic Editor and Responses

Academic Editor’s Comments:

1. Please ensure that your manuscript meets PLOS ONE's style requirements

Authors’ Response:

Dear Editor,

Thank you for the opportunity to submit our manuscript. We have meticulously ensured that our manuscript meets all PLOS ONE style requirements.

2. We noticed you have some minor occurrences of overlapping text with the following previous publication(s), which needs to be addressed

Authors’ Response:

Dear Editor,

We appreciate your attention to this matter and have promptly addressed and resolved the minor occurrences of overlapping text with the previous publication.

3. Please complete your Competing Interests on the online submission form to state any Competing Interests

Authors’ Response:

Dear Editor,

Thank you for the reminder. We have completed the Competing Interests section on the online submission form and included it with the cover letter.

4. We note that you have referenced (unpublished) on page 6. Please remove this from your References and amend this to state in the body of your manuscript: (ie “Bewick et al. [Unpublished]”) as detailed online in our guide for authors

Authors’ Response:

Dear Editor,

We apologize for the confusion caused. Upon review, we confirm that there is no unpublished citation on page six, and there is no reference named Bewick et al. in the manuscript. Sorry for any confusion this may have caused.

5. Please include captions for your Supporting Information files at the end of your manuscript

Authors’ Response:

Dear Editor,

Thank you for your recommendation. We have included captions for all Supporting Information files at the end of our manuscript as requested.

2. Comments from Reviewer #1 and Responses

Reviewer’s Comments:

1. Abstract: Add the number of studies and participants included in the review in the results section of the abstract.

Authors’ Response:

Dear Reviewer,

Thank you for your suggestion. We have revised the abstract to include the number of studies and participants included in the review as requested.

2. Introduction: NTDs has been spelled out more than once. Spell out once when used for the first time and then use the acronym for the rest of the manuscript.

Authors’ Response:

Dear Reviewer,

Thank you for your feedback. We have revised this in the manuscript and spelled out "NTDs" in the introduction once and used the acronym thereafter throughout the manuscript unless the term comes at the beginning of the statement.

3. Some of the references in the introduction are very old. I suggest replacing these with more recent references.

Authors’ Response:

Dear Reviewer,

Thank you for your suggestion. We replaced the older references in the introduction with more recent ones to ensure the currency of the information. Thank you once again for your valuable suggestion.

4. I found the introduction very weak. There are quite a few recent relevant references assessing the existing evidence, guidelines, and policies around folic acid supplementation. I suggest that the authors look at these papers and add some of the relevant references in their introduction to make a stronger case for the review

Authors’ Response:

Dear Reviewer,

Thank you for your suggestion. We appreciate the feedback and have reviewed recent relevant literature to strengthen the introduction by incorporating recent relevant references related to folic acid supplementation. The revised version of the manuscript incorporated the suggested updated articles and sources. This will enhance the robustness of our review.

5. Add some references (if available) on country-specific prevalence rates of NTDs in Africa.

Authors’ Response:

Dear Reviewer,

Thank you for the suggestion. We have incorporated recent relevant references on country-specific prevalence rates of neural tube defects in Africa to provide a more comprehensive introduction, aligning with existing evidence and guidelines on folic acid supplementation.

6. The acronym for sub-Saharan Africa (SSA), folic acid (FA), and low-middle-income countries (LMICs) have been used inconsistently throughout the manuscript. I would suggest that the authors spell it out when used for the first time in the draft and then consistently use the acronym throughout

Authors’ Response:

Dear Reviewer,

Thank you for your feedback. We have corrected it and ensured consistency in the use of acronyms such as SSA (Sub-Saharan Africa), FA (folic acid), and LMICs (low- and middle-income countries) throughout the manuscript.

7. Not sure what this sentence under 'study selection' means: "articles examining the prevalence and determinants of folic acid supplementation during the preconception period were gathered from various". From the objectives and results, it seems that only prevalence was assessed. I am not sure whether the authors have also looked at the determinants? and if they have, this has not been reported anywhere in the draft.

Authors’ Response:

Dear Reviewer,

We apologize for any confusion caused by the initial statement. The manuscript has been updated to accurately reflect our findings. The revised statement now reads:

"A comprehensive collection of articles assessing the prevalence of folic acid supplementation during the preconception period was gathered from various sources."

Thank you again.

8. Not sure what this sentence means: "A quantitative cross-sectional study design was employed to enhance clarity and significance." Do you mean that the eligibility for study selection was cross-sectional study design? If yes, then state this clearly.

Authors’ Response:

Dear Reviewer,

Thank you for your query regarding the statement "A quantitative cross-sectional study design was employed to enhance clarity and significance." We apologize for any confusion caused by this sentence. The intention of this statement was to clarify that our review specifically included studies employing quantitative methodologies, while excluding those using qualitative approaches. We revised the statement in the manuscript to explicitly state: “Quantitative studies were included to enhance clarity and significance.”

9. Suggest authors add a table or a figure depicting the regions and the respective countries included in Sub-Saharan Africa. It would add clarity for the readers as the regions and respective countries have been discussed and analyzed in the results and discussion sections

Authors’ Response:

Dear Reviewer,

Thank you for your valuable suggestion to add a table or figure depicting the regions and respective countries included in Sub-Saharan Africa. We have now included a table in the main manuscript under the characteristics of the studies section. This table outlines the regions of Sub-Saharan Africa and the respective countries within each region.

10. Study design is stated twice under data extraction

Authors’ Response:

Dear Reviewer,

Thank you for your observation. We have removed the duplicate mention of "study design" under the data extraction section.

11. Result: I can’t see the full characteristics of the included studies table (probably because it was in a landscape mode) so I am not sure if my comments are relevant. But I would suggest that the authors delete the columns on study period, center, sampling method, and response rate. I would also suggest that the authors add the columns on sample size and study results (reported prevalence) in the study characteristics table

Authors’ Response:

Dear Reviewer,

Thank you for your feedback regarding the table of included studies. We apologize if the landscape format made it difficult to view the entire table. Based on your suggestions, we made the following adjustments:

o We have deleted the columns on study period, center, sampling method, and response rate.

o We have added columns for sample size and study results (reported prevalence).

12. I would suggest that the authors remove the study quality column from the study characteristics table and add a separate section on study quality.

Authors’ Response:

Dear Reviewer,

Thank you for your feedback regarding the study characteristics table. We agree that separating the study quality information into its own section would enhance clarity. As such, we removed the study quality column from the study characteristics table and have prepared a separate section dedicated to study quality assessment. Thank you once again for your recommendation.

13. I would suggest that the authors re-organise the discussion section starting with summarising the reveiw findings, quality of the included studies, comparing the review findings with other studies/reviews, strengths and limitations of the review and research and practice implications.

Authors’ Response:

Dear Reviewer, Thank you for your valuable suggestion regarding the re-organization of the discussion section. We appreciate your feedback and agree that restructuring the section will enhance the clarity and coherence of our manuscript. In the discussion section, we summarized key findings, evaluated study quality, compared results with existing literature, and assessed methodological strengths and limitations. We concluded by discussing implications for future research and clinical practice.

14. I would suggest that the authors compare the study findings with studies conducted in similar context. For eaxmple, reference 59 is a study from Italy and hence widely differs in context. However, authors can broadly discuss the differences in the folic acid supplementation prevalance between high income and low- middle- income countries.

Authors’ Response:

Dear Reviewer, Thank you for your valuable feedback. We agree that Reference 59, a study from Italy, differs significantly in context compared to our study setting. To address this, we have broadened our discussion to compare folic acid supplementation prevalence between high-income and low- to middle-income countries. Although studies in this area are limited, we have made efforts to address this gap through a comprehensive search of existing literature. Thank you once again for your insightful suggestion.

3. Comments from Reviewer #2 and Responses

Reviewer’s Comments:

1. Introduction: Seems very long for an abstract. Just a suggestion: instead of reporting global estimates of NTDs I would suggest writing the estimates from Sub Saharan Africa. I dont think daily dosage is required here.

Authors’ Response:

 Dear Reviewer, Thank you for your feedback. We have focused on Sub-Saharan Africa estimates of NTDs and omitted daily dosage details to make the abstract more concise. Thank you for your suggestions.

2. Methods: Please add on what type of studies did you include? Last date of search? if GRADE was conducted and how was quality of included studies conducted (if you did any of these). 

Dear Reviewer, Thank you for your feedback. We have revised the methods section to include more details about the types of studies included, the last date of the search, and the quality assessment process using Newcastle-Ottawa Scale. Thank you for your suggestions.

3. Results: Line 44: not sure what do you mean by 'individual studies' here.

Line 46: subgroup analysis based on?

Line 49: remove double brackets from the end.

 quality of included studies in missing

Authors’ Response:

Thank you for your feedback.

Line 44: We apologize for any confusion caused. We have now clarified our statement.

Line 46: We conducted subgroup analysis based on corresponding countries.

Line 49: Thank you for your observation. We have removed the unnecessary information.

We employed the Newcastle-Ottawa Scale for quality assessment, as highlighted in the abstract and we have included it the main document.

Thank you once again for your careful review and for improving the abstract section.

4. Introduction If you have introduced the abbreviation once, then you dont have to introduce it again and again. e.g., NTDs. 

Authors’ Response:

Dear Reviewer, Thank you for your feedback. We have corrected it and ensured consistency in the use of abbreviations, such as NTDs, once introduced.

5. Methods: I would suggest to rearrange some headings. After registration, you should talk about the eligibility criteria and then search strategy and study selection and so on

Authors’ Response:

Dear Reviewer, Thank you for your suggestion. We have rearranged the headings to discuss eligibility criteria immediately following registration, followed by the search strategy, study selection, and subsequent sections. Thank you for guiding us in improving the organization of our methods section.

6. Line 117: you have stated that you search institutional repositories. Please state which were those?

Authors’ Response:

Dear Reviewer, Thank you for your comment. We have searched the institutional repositories of several universities, namely Addis Ababa University, Jimma University, Hawassa University, and Bahir Dar University. Thank you for your attention to this detail and your valuable feedback.

7. Line 120: Please check punctuation: "supplementation, and folic acid administration. folic acid adherence, folinic acid, and vitamin B9."

Authors’ Response:

Dear Reviewer, Thank you for your comment.

Line 120: We have corrected the punctuation as follows: “preconception care, preconception, pre-conception, periconception, peri-conception, before conception, folic acid, folate, vitamin, pteroylglutamic acid, folic acid supplementation, folic acid administration, folic acid adherence, folinic acid and vitamin B9”.Thank you for bringing this to our attention and for your careful review.

8. Line 127-129: This sentence is very vague "A quantitative cross-sectional study design was employed to enhance clarity and significance'

Authors’ Response:

Dear Reviewer,

Thank you for your feedback.

Line 127-129: We acknowledge the vagueness of the sentence "A quantitative cross-sectional study design was employed to enhance clarity and significance." We revised this statement to provide more specific details about our study design. Thank you for your thoughtful review and suggestions for improving clarity in our manuscript.

9. Line 127: 'omitted' is not an appropriate word to use here

Dear Reviewer, Thank you for your feedback. Line 127: We acknowledge that "omitted" is not appropriate in this context. We have revised the wording accordingly to accurately describe our methodology. Thank you for bringing this to our attention and for your careful review.

10. I think you need to revisit this paragraph. It requires language editing

Linme 127-129: " A quantitative cross-sectional study design was employed to enhance clarity and significance. In contrast, studies utilizing qualitative methodologies were omitted due to the specific nature of the review and the analytical approach selected for this review" I think this should be discussed under section 2.4.

Authors’ Response:

Dear Reviewer, Thank you for yo

---

## [Decision Letter · Decision Letter 1]

6 Sep 2024

PONE-D-24-13717R1Folic acid supplementation during preconception period in sub-Saharan African countries: A systematic review and meta-analysisPLOS ONE

Dear Dr. Aweke,

Thank you for submitting your manuscript to PLOS ONE. After careful consideration, we feel that it has merit but does not fully meet PLOS ONE’s publication criteria as it currently stands. Therefore, we invite you to submit a revised version of the manuscript that addresses the points raised during the review process.

We look forward to receiving your revised manuscript.

Kind regards,

Jai K Das

Academic Editor

PLOS ONE

Reviewers' comments:

Reviewer's Responses to Questions

**Comments to the Author**

1. If the authors have adequately addressed your comments raised in a previous round of review and you feel that this manuscript is now acceptable for publication, you may indicate that here to bypass the “Comments to the Author” section, enter your conflict of interest statement in the “Confidential to Editor” section, and submit your "Accept" recommendation.

Reviewer #1: (No Response)

Reviewer #2: (No Response)

2. Is the manuscript technically sound, and do the data support the conclusions?

Reviewer #1: Partly

Reviewer #2: Partly

3. Has the statistical analysis been performed appropriately and rigorously? 

Reviewer #1: Yes

Reviewer #2: Yes

4. Have the authors made all data underlying the findings in their manuscript fully available?

Reviewer #1: Yes

Reviewer #2: Yes

5. Is the manuscript presented in an intelligible fashion and written in standard English?

Reviewer #1: No

Reviewer #2: No

6. Review Comments to the Author

Reviewer #1: The review title and objectives suggest that the authors aimed to review the prevalance of folic acid supplementation during preconception period which is defined as the period of time before pregnancy occurs. However, the authors have used terms for 'periconception' as well in their search strategy. Moreover, the included studies had a variety of target population "eight studies (28.57%) were women of childbearing age, 16 studies 257 (57.1%) centered on pregnant mothers, and four studies (14.3%) included postnatal mothers." Majority of these studies (16 studies) included pregnant women and does not follow the eligibiltiy crirteria for inclusion which is preconception supplementation. The only studies eligible for inlcusion as per the authors eligibility criteria are studies that targeted either women of reproductive age or postnatal women. I would suggets that the authors clarify the definition of "preconception" used for this review to reflect the correct eligibility and inclusion.

Reviewer #2: Manuscript require thorugh read anf formatting edits. There are places were there is no space between text, references, and brackets e.g., Line 108: individuals(7).The ; Line 82: In sub-Saharan Africa(SSA).

Punctuation is not used appropriately e.g. like 127: .(28). - Remove extra full stops where evere required.

Please read your manuscript thoroughly and make the required edits where necessary.

Abstract:

Introduction is still very long. Please reduce some text

- Line 31: Remove extra brackets and full stop

- Line 40: Add space between of and FA

Line 152: This line is very confusing where you say that you are including ALL journal articles. However you are excluding qualitative studies. I get what you are trying to say here but its still very confusing. "We included all types of studies published as journal articles, theses, and dissertations without imposing restrictions based on publication date or age criteria."

Section 2.2 (Line 150): First write what was included and then write what was excluded. This brings more clarity.

line 149: Which quantitative studies did you include? please state. List all the study designs which you intended to included and in results you can state that you only found cross-sectional studies

Section 2.3:

- Line 164: "were taken" is not an appropriate work to use here

- In this section first talk about making the search strategy by using Mesh terms and key words. Then talk about databases searched

Section 2.4:

Line 176: Unclear

Line 177: "The search was not constrained by the publication year; hence, articles published until January 2024, were considered for review eligibility." You have already mentioned the last publication date in section 2.3. The remaining details on restrictions should be moved to section 2.3.

Line 180: The information is not making any sense to me. As 'systematic review" everything should be done systematically. Even contacting authors comes after data extraction. Not before screening. Please read some good published systematic reviews and suggest rewritting your methodology.

Line 186: This should come before title abstract screening.

Section 2.6 will come before 2.5

Section 2.6:

Did you summarise data in Excel or you conducted data extraction in excel? Please clarify

Did you cross-reference studies for missing studies?

Through NewCastle you assess the quality of included studies. Through GRADE you assess quality of evidence. both are two different things. However, I would suggest to use GRADE. Let us know if you dont have the capacity to do GRADE.

Results:

Line 231: "We reviewed the full texts of the remaining 195 articles for eligibility, 163 studies were not eligible, and 32 studies were identified." what do you mean by 32 were identified?

- Rephrase line 238

- Line 259: I think it should be table 2?

Please place all the tables appropriately.

Table 2: Suggest showing this in a graph and move table 2 to supplementary file.

Table 3: You said that you included pregnant women because they reterospectively collected data then how are those cross-secrional studies. wont those be reterospective cohort studies? Or these were cross-sectional who collected data based on recall. I just want to have an understanding on what type of studies were these.

- You are talking about sensitivity analysis in Section 3.5 and then the actual analysis comes in Section 3.7. I would suggest moving Section 3.7 to Section 3.5

- Line 357-360: There is no connection of discussing quality of studies of your review after discussing the intervention from existing evidence. Suggest moving it afetr line 341 and try to be more cohesive.

- Line 361: I think you discussion need more working. First, Talk about the evidence coming from you study, including the subgroup, sentivity and quality of studies. then compare with other studies conducted in Africa followed by studies conducted in LMIC and HIC countries. Then talk about stengths and limitations.

7. PLOS authors have the option to publish the peer review history of their article (what does this mean?). If published, this will include your full peer review and any attached files.

Reviewer #1: No

Reviewer #2: No

---

## [Author Response · Author response to Decision Letter 1]

20 Sep 2024

Academic editor’s comments Authors’ responses

Submit a revised version of the manuscript that addresses the points raised during the review process

Authors’ responses 

Dear editor,

Thank you for the opportunity to submit our manuscript.

we have thoroughly addressed and incorporated all of the reviewers' comments and suggestions into the revised version. We greatly appreciate the time and effort the editorial team have invested in reviewing our work, and we believe that these revisions have strengthened the manuscript.

2. Comments from reviewer #1 and responses

Reviewer’s comments 

The review title and objectives suggest that the authors aimed to review the prevalance of folic acid supplementation during preconception period which is defined as the period of time before pregnancy occurs. However, the authors have used terms for 'periconception' as well in their search strategy. Moreover, the included studies had a variety of target population "eight studies (28.57%) were women of childbearing age, 16 studies 257 (57.1%) centered on pregnant mothers, and four studies (14.3%) included postnatal mothers." Majority of these studies (16 studies) included pregnant women and does not follow the eligibiltiy crirteria for inclusion which is preconception supplementation. The only studies eligible for inlcusion as per the authors eligibility criteria are studies that targeted either women of reproductive age or postnatal women. I would suggets that the authors clarify the definition of "preconception" used for this review to reflect the correct eligibility and inclusion. 

Authors’ responses

Dear reviewer, 

Thank you for your valuable feedback We understand your concern regarding the search terms we used and the target populations of the included studies. 

We used the term "periconception" in our search strategy because some studies report folic acid supplementation during the preconception period separately, even though the studies were conducted during the periconception phase. By including "periconception," we aimed to ensure that we captured all relevant studies, particularly those that might have reported preconceptional folic acid intake under the broader term "periconception.” So, we specifically selected only those that reported on folic acid supplementation during the preconceptional(before pregenency) period.

Regarding the inclusion of studies involving pregnant, we included studies with pregnant participants because they assessed folic acid (FA) intake during the preconception period by asking women about their supplementation practices before the pregnancy based on recall. Even though these studies involve pregnant women at the time of the survey, they provided valuable information about preconceptional FA intake. However, we did not include studies that reported FA supplementation during pregnancy. Our aim was to focus specifically on preconceptional FA supplementation to ensure the relevance of the data.

Thank you for your detailed review and for highlighting these points. We appreciate your insights and hope this clarifies our approach.

3. Comments of reviewer #2 and responses

Reviewer’s comments 

Manuscript require thorugh read anf formatting edits. There are places were there is no space between text, references, and brackets e.g., Line 108: individuals(7).The ; Line 82: In sub-Saharan Africa(SSA). Punctuation is not used appropriately e.g. like 127: .(28). - Remove extra full stops where evere required.

Please read your manuscript thoroughly and make the required edits where necessary. 

Authors’ responses

Dear reviewer,

Thank you for your feedback. We have reviewed and corrected the formatting issues, including spacing between text, references, and brackets, and adjust punctuation errors such as extra full stops.

We appreciate your assistance and will make the necessary edits.

Reviewer’s comments 

Abstract:

Introduction is still very long. Please reduce some text

- Line 31: Remove extra brackets and full stop

- Line 40: Add space between of and FA

Authors’ responses

Dear reviewer,

Thank you for your carful review and your relevant comments. According to the suggestions we reduced the texts from the introduction of abstract section and we have corrected the errors of and we removed in appropriate punctuations. Thanks again for your suggestions.

Line 152: This line is very confusing where you say that you are including ALL journal articles. However you are excluding qualitative studies. I get what you are trying to say here but its still very confusing. "We included all types of studies published as journal articles, theses, and dissertations without imposing restrictions based on publication date or age criteria." Dear reviewer,

Thank you for requesting the clarity of the confusing stetment. We have rewrite the statement as the following and we hope it is clear.

“We included journal articles, theses, and dissertations without imposing restrictions based on publication date or age criteria.”

Reviewer’s comments 

Section 2.2 (Line 150): First write what was included and then write what was excluded. This brings more clarity.

line 149: Which quantitative studies did you include? please state. List all the study designs which you intended to included and in results you can state that you only found cross-sectional studies. 

Authors’ responses

Dear reviewer.

Thank you for your valuable comments and suggestions. We have corrected the order of the statements, first specifying the included studies and then the excluded ones. We included cross-sectional, cohort, and case-control studies, but ultimately found only cross-sectional studies in our review. 

We mentioned the included study designs in the main document.

Thank you again for your efforts to improve the manuscript.

Reviewer’s comments 

Section 2.3:

- Line 164: "were taken" is not an appropriate work to use here

- In this section first talk about making the search strategy by using Mesh terms and key words. Then talk about databases searched.

Authors’ responses

Dear reviewer,

Thank you for your comment. We have written the stetment according to the comment as the following.

“Studies published in the English language up to January 2024 were retrieved from EMBASE, MEDLINE, Scopus, CINAHL, and manually on Google and Google Scholar.”

In addition in this section first we mentioned about mesh terms and key words then we explained about data base search. Thank you for again for your important comments.

Reviewer’s comments 

Section 2.4:

Line 176: Unclear

Authors’ responses

Dear Reviewer,

Thank you for your comment.

We apologize for the confusion caused by the unclear statement. We have corrected it in the main document as follows:

“We systematically reviewed and evaluated studies from various sources to ensure a comprehensive analysis.”

Reviewer’s comments 

Line 177: "The search was not constrained by the publication year; hence, articles published until January 2024, were considered for review eligibility." You have already mentioned the last publication date in section 2.3. The remaining details on restrictions should be moved to section 2.3. 

Authors’ responses

Dear reviewer, 

Thank you for your comment. We agree that we have already explained this under section 2.3 and we have removed the statement from Line 177, as the details about publication year constraints are already explained in Section 2.3.

Reviewer’s comments 

Line 180: The information is not making any sense to me. As 'systematic review" everything should be done systematically. Even contacting authors comes after data extraction. Not before screening. Please read some good published systematic reviews and suggest rewritting your methodology.

Authors’ responses

Dear reviewer, 

Thank you for your feedback. We acknowledge that contacting authors should occur after data extraction in a systematic review. We have corrected the arrangement of stetments and we mentioned contacting Authors for missed data were under the data extraction section. 

Reviewer’s comments 

Line 186: This should come before title abstract screening

Authors’ responses

Dear reviewer, 

Thank you for your valuable suggestion. To maintain the sequence of the review process, we moved the statement from Line 86 to precede the abstract and title screening stetments.

Reviewer’s comments 

Section 2.6 will come before 2.5 

Authors’ responses

Dear reviewer,

Thank you for your feedback. We have revised the manuscript to present the section 2.6 before section 2.5. 

Thank you again for your relevant suggestions.

Reviewer’s comments 

Section 2.6:

Did you summarise data in Excel or you conducted data extraction in excel? Please clarify

Authors’ responses

Dear reviewer, 

Thank you for your question. We apologize for the confusion that cuase in the writing this process. We conducted the data extraction using Microsoft Office Excel and we clarify it in the main document this process.

Reviewer’s comments 

Did you cross-reference studies for missing studies? 

Authors’ responses

Dear reviewer. 

Thank you for the feedback. We have cross-referenced the included studies to ensure no relevant studies were missed.

Reviewer’s comments 

Through NewCastle you assess the quality of included studies. Through GRADE you assess quality of evidence. both are two different things. However, I would suggest to use GRADE. Let us know if you dont have the capacity to do GRADE. 

Authors’ responses

Dear reviewer,

Thank you for your valuable feedback. We have carefully assessed the certainty of evidence using the GRADE approach, as requested. The evaluation was conducted independently by two authors (E.A. and M.C.), with any disagreements resolved by consensus. This process has been clearly documented in the revised manuscript. We appreciate your suggestion, which has helped improve the clarity and transparency of our study.

Reviewer’s comments 

Results:

Line 231: "We reviewed the full texts of the remaining 195 articles for eligibility, 163 studies were not eligible, and 32 studies were identified." what do you mean by 32 were identified? 

Authors’ responses

Dear Reviewer, 

Thank You for your request regarding to the clarity of the stetment. We corrected the stetment in the main document as the following;

“We reviewed the full texts of the remaining 195 articles for eligibility. Of these, 163 studies were not eligible, and 32 studies were eligible for inclusion in our review”

Reviewer’s comments 

Rephrase line 238 

Authors’ responses

Dear reviewer,

Thank You for your suggestion. We rephrased the statement as the following;

“We excluded three studies from the analysis: two conducted in Ethiopia(36, 37) and one in Uganda(38), because they reported a prevalence of folic acid supplementation during the preconception period as zero percent (0%).”

Reviewer’s comments 

- Line 259: I think it should be table 2?

Please place all the tables appropriately.

Table 2: Suggest showing this in a graph and move table 2 to supplementary file. 

Authors’ responses

Dear Reviewer,

Thank you for your suggestion. We have moved Table 2, which covers the quality of the study, to the supplementary file and placed the other tables in their appropriate locations as needed. We appreciate your guidance.

Reviewer’s comments 

Table 3: You said that you included pregnant women because they reterospectively collected data then how are those cross-secrional studies. wont those be reterospective cohort studies? Or these were cross-sectional who collected data based on recall. I just want to have an understanding on what type of studies were these. Authors’ responses

Dear reviewer,

Thank you for your clarification request. The studies were cross-sectional whose target population were pregnant women. However, this study collected data by asking pregnant women to recall and report their folic acid (FA) supplementation history from before they became pregnant.

Reviewer’s comments 

You are talking about sensitivity analysis in Section 3.5 and then the actual analysis comes in Section 3.7. I would suggest moving Section 3.7 to Section 3.5

Authors’ responses

Dear reviewer,

Thank you for pointing that out. To improve the flow, we have moved Section 3.7 to Section 3.5, aligning the sensitivity analysis with its results for improved coherence.

Reviewer’s comments 

- Line 357-360: There is no connection of discussing quality of studies of your review after discussing the intervention from existing evidence. Suggest moving it afetr line 341 and try to be more cohesive. 

Authors’ responses

Dear reviewer,

Thank you for the suggestion. We moved the discussion on study quality to follow after line 341 to enhance coherence and ensure a better connection with the preceding content on the intervention.

Reviewer’s comments 

- Line 361: I think you discussion need more working. First, Talk about the evidence coming from you study, including the subgroup, sentivity and quality of studies. then compare with other studies conducted in Africa followed by studies conducted in LMIC and HIC countries. Then talk about strengths and limitations. 

Authors’ responses

Dear reviewers, 

Thank you for your valuable suggestions. We have thoroughly revised the discussion section. We have incorporated evidence from our study, including subgroup analysis, sensitivity, and study quality, followed by comparisons with African studies and research from LMICs and HICs. We have also addressed the strengths and limitations of our work. We appreciate your helpful comments and feedback.

We appreciate the reviewer's and editor’s thoughtful comments and suggestions, which have significantly contributed to enhancing the clarity and robustness of our manuscript.

---

## [Decision Letter · Decision Letter 2]

11 Nov 2024

PONE-D-24-13717R2Folic acid supplementation during preconception period in sub-Saharan African countries: A systematic review and meta-analysisPLOS ONE

Dear Dr. Aweke,

Thank you for submitting your manuscript to PLOS ONE. After careful consideration, we feel that it has merit but does not fully meet PLOS ONE’s publication criteria as it currently stands. Therefore, we invite you to submit a revised version of the manuscript that addresses the points raised during the review process.

We look forward to receiving your revised manuscript.

Kind regards,

Jai K Das

Academic Editor

PLOS ONE

Journal Requirements:

**Additional Editor Comments:**

I would like to thank the authors for conducting this systematic review and thoroughly revising the manuscript based on the feedback from the reviewers. I have some additional concerns which are stated below:

Please specify that in the methods and results that studies done on pregnant women were only included if they had taken folate during preconception and also clarify that the studies only included these women or was there a mix.

I would suggest not to use the word ‘prevalence’ and rather use ‘proportion’ as there were studies included which were conducted in facilities and did not have a sampling frame to assess prevalence and even a few community studies would have done so. The authors need to define whether the studies covered few geographic locations or were nation level.

The introduction is lengthy and should be shortened

In the results and the PRISMA diagram – specify that how many studies were included from grey literature

Specify how many authors were contacted for missing information

Why two studies reporting zero prevalence were excluded from the analysis as they have reported the proportion, and this can add to the meta-analysis. If there is any specific reason, then the authors should state that.

In the table of included studies – please also specify the scale of the included studies – whether from one village, district, province or national and if from facilities – then how many facilities were covered.

Can the authors add a sub-group by urban/rural

Please thoroughly check the manuscript for language and grammar

Reviewers' comments:

Reviewer's Responses to Questions

**Comments to the Author**

1. If the authors have adequately addressed your comments raised in a previous round of review and you feel that this manuscript is now acceptable for publication, you may indicate that here to bypass the “Comments to the Author” section, enter your conflict of interest statement in the “Confidential to Editor” section, and submit your "Accept" recommendation.

Reviewer #1: (No Response)

Reviewer #2: All comments have been addressed

2. Is the manuscript technically sound, and do the data support the conclusions?

Reviewer #1: Yes

Reviewer #2: Yes

3. Has the statistical analysis been performed appropriately and rigorously? 

Reviewer #1: Yes

Reviewer #2: Yes

4. Have the authors made all data underlying the findings in their manuscript fully available?

Reviewer #1: Yes

Reviewer #2: Yes

5. Is the manuscript presented in an intelligible fashion and written in standard English?

Reviewer #1: Yes

Reviewer #2: Yes

6. Review Comments to the Author

Reviewer #1: The authors have responded to the concerns raised over the last two rounds of review. I am happy with the editors recommendations if authors can make the edits to the eligibility criteria specifying that only data from studies where outcomes were specific to preconception supplementation were included. As the readers might get a bit confused looking at the 'participants' column in the characteristics studies.

Reviewer #2: Minor:

Table 1: add brackets around C after Comparison

Line 188: remove extra full stops.

Line 187: 'data' should be in lowercase

Check spacing: sampling method(random/non-random)

Table 5: I did inquire initially about using Gradepro for GRADE. I would suggest giving low/moderate/serious etc ranking for each domain and providing the explanation of each ranking as footnotes.

7. PLOS authors have the option to publish the peer review history of their article (what does this mean?). If published, this will include your full peer review and any attached files.

Reviewer #1: No

Reviewer #2: No

---

## [Author Response · Author response to Decision Letter 2]

19 Nov 2024

Date: Nov 13, 2024

To: Plos One, Editorial Office

Subject: Submitting a revised version of manuscript and point by point response

Manuscript ID: PONE-D-24-13717R1

I am pleased to resubmit our revised manuscript titled, "Folic acid supplementation during preconception period in sub-Saharan African countries: A systematic review and meta-analysis" (ID: PONE-D-24-13717R1). We have carefully considered and addressed the reviewers' valuable comments, making revisions that we believe have strengthened the clarity and quality of the manuscript.

This work highlights an important public health issue, and we are grateful for the opportunity to improve it based on the feedback provided. We hope it now meets the high standards of PLOS ONE and look forward to your assessment. Thank you for your time and consideration.

Regards,

Authors

Point by point response to the Academic editor and reviewer’s comments 

1. Comments from the academic editor and responses

Response 

Dear editor,

Thank you for your helpful feedback. Upon carefully reviewing the reference list, we found that two issues needed to be address the following citation was mistakenly cited in the manuscript:

Gbemileke A, Atta AO, Abolade T, Adegoke F, Arogundade E, Kung'u J, Beckworth C. Assessing women's knowledge on benefits of iron and folic acid and its consumption during pregnancy in northern Nigeria. Annals of Nutrition and Metabolism. 2023;79:636.

we have correct reference that should have been included is:

‘Adebo OO, Dairo DM, Ndikom CM, Adejumo PO. Knowledge and uptake of folic acid among pregnant women attending a secondary health facility in Nigeria. British Journal of Midwifery. 2017 Jun 2;25(6):358-64.’ 

This reference is accurate and has now been included in the updated reference list. Thank you for your understanding and continued guidance.

Editor comment

Please specify that in the method results studies done one pregenent women were only included if they had taken folate during preconception and clarify that the studies only included these women or were there a mix.

Authors response

Dear Editor, 

Thank you for your clarification question and recommendation to clarify this clearly in the manuscript. We have only included studies with pregenent women participants when they were take folate before pregnency/during preconception period.

Editors request

I would suggest not to use ‘prevalence’ and rather use ‘proportion’ as there were studies included which were conducted in facilities and did not have a sampling frame to assess prevalence and even few community studies would have done so. The authors need to define weather the studies covered few geographic locations or were national level.

Authors response

Dear editor,

Thank you for your suggestion to use proportion rather than prevalence due to the nature of the included studies. we have accepted and using proportion is more convenient than proportion. So we have corrected it accordingly. Thank you again for your valuable suggestion. 

Editor request

In the result and prisma diagram specify how many studies were included from grey litrature 

Authors response

Dear Editor,

Thank you for your request to specify the number of grey literatures in PRISMA flow diagram.

We have specified the number of grey literatures included to this SRMA study in the PRISMA flow diagram. 

Thank you again for your valuable comments.

Editors request

The introduction is lengthy and to be shortened

Authors response

Dear Editor,

Thank you for your recommendation to shorten the introduction section. We have made it shorten with removing unnecessary words and phrases to make stronger and more informative introduction. Thank you again for your suggestion

editors request

Why two studies zero prevalence were excluded from the analysis as they have reported the proportion and this can add to the meta-analysis. if there is specific reason the authors should state that. 

Authors response

Dear Editor 

Thank you for your concern and request regarding to exclusion zero prevalence studies. 

We excluded studies reporting zero prevalence from the analysis because, after closer reviewing them, we found that folic acid (FA) intake wasn’t the main focus of these studies. While they reported zero prevalence for FA intake, this wasn’t a central finding for their research, which made us concerned about how relevant and reliable that data would be for our meta-analysis. Additionally, the sample size and the methods used in these studies could have led to inaccurate reporting of zero prevalence. Including them could have introduced bias and affected the overall results, so we decided to exclude them to ensure the accuracy and reliability of our findings.

Editors request

In the table of included studies – please also specify the scale of the included studies – whether from one village, district, province or national and if from facilities – then how many facilities were covered

Authors response

Dear Editor,

Thank you for your insightful suggestion to specify the scale/level of the included studies. We have now clarified the scale/level for each study, and for those conducted in health facilities, we have included the number of facilities in the table.

Once again, we appreciate your valuable feedback.

Editors request

Can the authors add a sub-group by urban/rural

Authors response

Dear Editor,

Thank you for your suggestion to conduct a sub-group analysis by urban/rural. However, since most of the studies were conducted in health facilities with participants from both urban and rural areas, we were unable to perform the sub-group analysis. 

Editors request

Please thoroughly check the manuscript for language and grammar

Authors response

Dear Editor,

Thank you for your suggestion to review the manuscript for language and grammar errors. We have carefully reviewed and corrected the manuscript and remain open to any additional suggestions or comments regarding any missed errors.

2. Comments from reviewer #1 and responses

Reviewer’s comments

Reviewer #1: The authors have responded to the concerns raised over the last two rounds of review. I am happy with the editors recommendations if authors can make the edits to the eligibility criteria specifying that only data from studies where outcomes were specific to preconception supplementation were included. As the readers might get a bit confused looking at the 'participants' column in the characteristics studies.

Authors’ responses

Dear Reviewer,

Thank you for your crucial and helpful comments throughout the review process. Regarding the study participants, we have clarified that all the included studies focused on women were included only if women took folate before pregnancy. Thank you again for your valuable feedback.

3. Comments from reviewer #2 and responses

Table 1: add brackets around C after Comparison

Line 188: remove extra full stops.

Line 187: 'data' should be in lowercase

Check spacing: sampling method(random/non-random)

 Authors' response

Dear Reviewer,

Thank you for your comments and suggestions throughout the review process. Based on your feedback, we have corrected the typographical errors by removing extra full stops, adding brackets, changing the "D" in "data" to lowercase, and removing double spacing. We appreciate your valuable comments and suggestions.

Reviewers comment

Table 5: I did inquire initially about using Gradepro for GRADE. I would suggest giving low/moderate/serious etc ranking for each domain and providing the explanation of each ranking as footnotes.

Authors response

Dear reviewer,

Thank you for your inquiry. Based on your recommendation and the GRADE assessment criteria, we have revised the table and added footnotes with explanation of each ranking. 

We appreciate the reviewer's and editor’s thoughtful comments and suggestions, which have significantly contributed to enhancing the clarity and robustness of our manuscript.

---

## [Editor Report · Decision Letter 3]

22 Dec 2024

PONE-D-24-13717R3Folic acid supplementation during preconception period in sub-Saharan African countries: A systematic review and meta-analysisPLOS ONE

Dear Dr. Aweke,

Thank you for submitting your manuscript to PLOS ONE. After careful consideration, we feel that it has merit but does not fully meet PLOS ONE’s publication criteria as it currently stands. Therefore, we invite you to submit a revised version of the manuscript that addresses the points raised during the review process.

We look forward to receiving your revised manuscript.

Kind regards,

Jai K Das

Academic Editor

PLOS ONE

Journal Requirements:

**Additional Editor Comments:**

i would like to thank the authors for the revisions. I have one remaining comment about the three studies reporting zero prevalence being excluded from the analysis.

The authors should either exclude these studies altogether from the review or give the reason in the manuscript for not including them in the meta-analysis and if folic acid (FA) intake wasn’t the main focus of these studies, then why were they included. Also I am not clear how the authors assessed if the sample size and the methods used in these studies were not accurate (as the Newcastle-Ottawa Scale ratings do not suggest so) and how was this ascertained for all the studies that were included. i did have a look at it and did not see any compelling issues as preconception care was assessed in these studies and IFA was one component assessed. Please elaborate on this.

---

## [Author Response · Author response to Decision Letter 3]

24 Dec 2024

Date: December 24, 2024

To: PLOS ONE Editorial Office

Subject: Resubmission of Revised Manuscript (ID: PONE-D-24-13717R1)

Dear Editors,

We are pleased to resubmit our revised manuscript, "Folic Acid Supplementation During the Preconception Period in Sub-Saharan African Countries: A Systematic Review and Meta-Analysis" (ID: PONE-D-24-13717R1). We have carefully addressed the editor's comments and made revisions to enhance the manuscript's clarity and quality. Thank you for the opportunity to improve our work, and we look forward to your assessment.

Sincerely,

The Authors

Authors response: Thank you for your feedback. We have reviewed the references and replaced the incomplete citation, "Organization WH. Global health estimates (GHE)–Cause-specific mortality. 2015. 2015," with the complete and relevant reference:

Flores A, Vellozzi C, Valencia D, Sniezek J. Global burden of neural tube defects, risk factors, and prevention. Indian Journal of Community Health. 2014;26(Suppl 1):3.

We also replaced

World Health Organization. Prevention of neural tube defects. Department of Making Pregnancy Safer Switzerland: World Health Organization. 2006

with the more relevant and complete citation:

Gomes S, Lopes C, Pinto E. Folate and folic acid in the periconceptional period: recommendations from official health organizations in thirty-six countries worldwide and WHO. Public Health Nutrition. 2016 Jan;19(1):176-89.

We have replaced the incomplete reference:

The GRADE handbook H. Schünemann, J. Brożek, G. Guyatt and A. Oxman Cochrane Collaboration London, UK 2013

with the complete and updated citation:

Guyatt GH, Thorlund K, Oxman AD, Walter SD, Patrick D, Furukawa TA, Johnston BC, Karanicolas P, Akl EA, Vist G, Kunz R. GRADE guidelines: 13. Preparing summary of findings tables and evidence profiles—continuous outcomes. Journal of Clinical Epidemiology. 2013 Feb 1;66(2):173-83.

While we have ensured accuracy, we will accept if any incomplete references remain that may make tracking difficult. 

Academic editor’s comments:i would like to thank the authors for the revisions. I have one remaining comment about the three studies reporting zero prevalence being excluded from the analysis.

The authors should either exclude these studies altogether from the review or give the reason in the manuscript for not including them in the meta-analysis and if folic acid (FA) intake wasn’t the main focus of these studies, then why were they included. Also I am not clear how the authors assessed if the sample size and the methods used in these studies were not accurate (as the Newcastle-Ottawa Scale ratings do not suggest so) and how was this ascertained for all the studies that were included. i did have a look at it and did not see any compelling issues as preconception care was assessed in these studies and IFA was one component assessed. Please elaborate on this.

Authors response: Dear Editor,

Thank you for your insightful feedback and for pointing out this issue. We excluded the three studies with zero prevalence. These studies reporting zero prevalence were excluded from the review because their primary focus was not on folic acid (FA) intake during the preconception period. While these studies report zero prevalnce during preconception, FA supplementation was only a minor component, which could significantly influence the results and potentially compromise the reliability of the review findings.

Excluding these studies from review strengthens the specificity and reliability of our systematic review and meta-analysis in estimating FA supplementation during the preconception period. We have updated the manuscript accordingly and appreciate your guidance in helping us improve the quality of our work.

Thank you once again for your support to enhance the manuscript.

---

## [Editor Report · Decision Letter 4]

30 Dec 2024

PONE-D-24-13717R4Folic acid supplementation during preconception period in sub-Saharan African countries: A systematic review and meta-analysisPLOS ONE

Dear Dr. Aweke,

Thank you for submitting your manuscript to PLOS ONE. After careful consideration, we feel that it has merit but does not fully meet PLOS ONE’s publication criteria as it currently stands. Therefore, we invite you to submit a revised version of the manuscript that addresses the points raised during the review process. I would like to thank the authors for all the hard work and making the revisions and addressing most comments but I taking am still not clear on why the three studies with zero-prevalence are being excluded as the reasons that 'primary focus was not on folic acid (FA)' and a 'minor component' are not strong enough reasons as previously they were included in the review though not in meta-analysis. Also the exclusion criteria of your review does not specify exclusion on these reasons. The quality assessment done by the authors for the three studies is also not very different from the other studies and especially the domain of '**Assessment of outcome'.** So I would suggest to include these three or provide a strong reason (within the pre-set exclusion criteria) and how it varies from all the other included studies.     

We look forward to receiving your revised manuscript.

Kind regards,

Jai K Das

Academic Editor

PLOS ONE
---

## [Author Response · Author response to Decision Letter 4]

1 Jan 2025

To: PLOS ONE Editorial Office

Subject: Resubmission of Revised Manuscript (ID: PONE-D-24-13717R1)

Dear Editors,

We are pleased to resubmit our revised manuscript, "Folic Acid Supplementation During the Preconception Period in Sub-Saharan African Countries: A Systematic Review and Meta-Analysis" (ID: PONE-D-24-13717R1). We have carefully addressed the editor's comments and made revisions to enhance the manuscript's clarity and quality. Thank you for the opportunity to improve our work, and we look forward to your assessment.

Sincerely,

The Authors

Academic editor’s comments

I would like to thank the authors for all the hard work and making the revisions and addressing most comments but I taking am still not clear on why the three studies with zero-prevalence are being excluded as the reasons that 'primary focus was not on folic acid (FA)' and a 'minor component' are not strong enough reasons as previously they were included in the review though not in meta-analysis. Also the exclusion criteria of your review does not specify exclusion on these reasons. The quality assessment done by the authors for the three studies is also not very different from the other studies and especially the domain of 'Assessment of outcome'. So I would suggest to include these three or provide a strong reason (within the pre-set exclusion criteria) and how it varies from all the other included studies. 

Authors’ responses

Dear Editors We appreciate your thoughtful comments and suggestion. The decision to exclude these three studies was not only 'primary focus was not on folic acid (FA)' and a 'minor component but also based on the fact that their results were extremely low, reporting zero values, which are considered outliers in this context. 

Including these studies in the meta-analysis would not allow for a standard pooled estimate, as zero values lead to a negative standard deviation, which is not acceptable. To resolve this, a continuity correction formula would need to be applied, but doing so would still introduce bias by modifying the numbers inappropriately. Therefore, to maintain the robustness and integrity of the study, we decided to exclude these studies. We have added the exclusion criteria to the Methods section for greater clarity and transparency. However, we remain open to your suggestions and are happy to reconsider the approach if necessary. Thank you again for your insightful comments and suggestions.

Journal Requirements:

Authors’ responses

We have reviewed the reference list, and none of the cited papers have been retracted. All references are complete and correct

---

## [Editor Report · Decision Letter 5]

16 Jan 2025

Folic acid supplementation during preconception period in sub-Saharan African countries: A systematic review and meta-analysis

PONE-D-24-13717R5

Dear Dr. Aweke,

We’re pleased to inform you that your manuscript has been judged scientifically suitable for publication and will be formally accepted for publication once it meets all outstanding technical requirements.

Kind regards,

Jai K Das

Academic Editor

PLOS ONE
---

## [Editor Report · Acceptance letter]

18 Jan 2025

PONE-D-24-13717R5 

PLOS ONE

Dear Dr. Aweke, 

I'm pleased to inform you that your manuscript has been deemed suitable for publication in PLOS ONE. Congratulations! Your manuscript is now being handed over to our production team.

Kind regards, 

on behalf of

Dr. Jai K Das 

Academic Editor

PLOS ONE
